# Strong intermodel differences and biases in CMIP6 simulations of PM$_{2.5}$, aerosol optical depth, and precipitation over Africa

**Catherine A. Toolan**[1], **Joe Adabouk Amooli**[2], **Laura J. Wilcox**[3], **Bjørn H. Samset**[4], **Andrew G. Turner**[1,3], **and Daniel M. Westervelt**[2,5]

[1]Department of Meteorology, University of Reading, Reading, United Kingdom
[2]Lamont-Doherty Earth Observatory of Columbia University, Palisades, NY, United States of America
[3]National Centre for Atmospheric Science, University of Reading, Reading, United Kingdom
[4]Center for International Climate and Environmental Research (CICERO), Oslo, Norway
[5]NASA Goddard Institute for Space Studies, New York, NY, United States of America

**Correspondence:** Catherine A. Toolan (c.toolan@pgr.reading.ac.uk)

**Abstract.** Poor air quality and precipitation change are strong, rapidly changing, and possibly linked drivers of physical hazards in sub-Saharan Africa. Future projections of sub-Saharan air quality and precipitation remain uncertain due to differences in model representations of aerosol, aerosol–precipitation interactions, and unclear future aerosol emission pathways. In this study, we evaluate the performance of CMIP6 models in simulating PM$_{2.5}$, aerosol optical depth (AOD), and precipitation over Africa relative to a range of observational and reanalysis products, including novel observational datasets, over the 1981–2023 period. While models accurately capture the seasonal cycle of PM$_{2.5}$ concentrations over most regions, the concentration magnitudes show strong intermodel diversity. Dust AOD shows a generally accurate seasonal spatial distribution, with multi-model mean (MMM) pattern correlation coefficients within 0.77–0.94, despite strong intermodel diversity in magnitude. Seasonal spatial patterns of non-dust AOD are poorly represented, with MMM pattern correlation coefficients of 0.25–0.58 and the poorest performance during September through November. Emission inventory inaccuracies may explain systematic biases for non-dust AOD fields, with differences in circulation and precipitation patterns, as well as aerosol treatment causing intermodel diversity. The magnitude and annual progression of precipitation over both the east and west African monsoon regions are well captured, though there is poorer performance in simulating the east African monsoon. Biases found relate to the intertropical convergence zone, more apparent over east Africa, and rainfall magnitude, more apparent over west Africa. This evaluation highlights strong intermodel diversity in the representation of African air quality and climate and identifies model performance over sub-Saharan Africa and the reasons behind the biases as critical gaps to address for improving confidence in climate projections.

## 1 Introduction

Africa is a region of large spatial heterogeneity in both air quality and precipitation (Hulme, 2001; Bauer et al., 2019). Variations in air quality are not well characterised due to a scarcity of long-term observations (United Nations Environment Programme, 2017). However, initial studies of individual regions in Africa (Kalisa et al., 2023; Kebede et al., 2021) have shown high spatial and temporal variability on interannual and multidecadal timescales. Precipitation variability is manifested in severe drought conditions and floods, often affecting the same region from year to year (Lüdecke et al., 2021).

The variability of air quality and precipitation both have strong impacts on public health. For example, air pollution was the second-highest risk factor for mortality across Africa in 2019 (Health Effects Institute, 2022; Xing et al., 2016), placed above that associated with unsafe water, sanitation, and hygiene. Africa exhibits high levels of PM$_{2.5}$ (McFarlane et al., 2021; Raheja et al., 2023; Westervelt et al., 2023) – aerosols with an aerodynamic diameter of less than 2.5 μm (Seinfeld and Pandis, 2016). High concentrations of PM$_{2.5}$ are a known cause of increased morbidity and mortality (Xing et al., 2016), so capturing the evolution of PM$_{2.5}$ is essential for mitigation strategies. The second leading cause of illness burden in most of sub-Saharan Africa is household air pollution from solid fuels, which adds to ambient particulate matter pollution (Katoto et al., 2019). Air quality is undergoing changes, with a continent-wide increase in aerosol emissions associated with population growth, urbanisation, and industrialisation (Wei et al., 2021), demonstrated by increasing aerosol optical depth (AOD) and higher PM$_{2.5}$ levels over the majority of Africa (Turnock et al., 2020). Africa has undergone, and continues to undergo, strong variations in annual precipitation. Drought in the Sahel from 1968 to 1973 was marked by a sudden shift from the wetter regime of the 1960s to a multi-year dry anomaly (Hulme, 2001). The drought was found to be directly linked to an estimated 100 000 deaths (Copans, 2019), as well as leading to the loss of 40 %–60 % of livestock in the region (Glantz, 1976). While annual mean precipitation has steadily recovered since 1983 in the western Sahel (Porkka et al., 2021), the eastern coast of Africa has experienced further drought conditions, with the Greater Horn of Africa undergoing its most severe drought in 40 years between 2018 and 2023 (World Health Organisation, 2024).

PM$_{2.5}$, AOD, and precipitation all exhibit inter-linkages at different spatial scales. PM$_{2.5}$ concentrations relate to the large-scale behaviour of AOD and precipitation patterns, as well as local aerosol emissions. AOD and PM$_{2.5}$ concentrations are closely linked, as both are measures of particulate matter in the atmosphere. However, the variables do not necessarily align spatially with each other, due to differences in sources and compositional differences, and the fact that PM$_{2.5}$ is generally measured at the surface, whereas AOD is a whole-column measurement that also takes aerosol radiative properties into account.

Absorbing and scattering aerosols also strongly affect precipitation, through changes to the local energy balance and lapse rates (Williams et al., 2023; Samset, 2022), and induce radiative forcing directly through the extinction of incoming solar radiation and indirectly through the modification of cloud microphysical properties (Boucher et al., 2013; Samset et al., 2018; Westervelt et al., 2020). Enhanced aerosol levels increase cloud condensation nuclei (CCN) concentrations (Twomey et al., 1984), resulting in more numerous but smaller cloud droplets, which increases cloud albedo and potentially increases cloud lifetime (Twomey, 1977) and can affect the efficiency of precipitation formation and the size distribution of raindrops (Westervelt et al., 2017; Gupta et al., 2023; Stier et al., 2024; Levin and Cotton, 2008). In addition, temperature profile modifications through heating due to absorbing aerosols alter atmospheric stability, changing cloud cover and therefore impacting the Earth's albedo (Ramanathan and Carmichael, 2008). The strongly regional forcing due to aerosol changes leads to changes in the atmospheric circulation, both locally and remotely. Therefore, the representation of aerosol processes is integral to capturing regional changes from anthropogenic forcing.

Precipitation also impacts air pollution, causing removal of aerosols from the atmosphere via wet deposition (Fuzzi et al., 2015; Wang et al., 2023) and influencing dust emissions via soil moisture, so changes in precipitation can cause feedbacks. Thus, it can be expected that relationships exist between African air quality and precipitation, though the response of precipitation to changes in aerosol emissions is dependent on the background state of both the emission and response region (Persad, 2023). From the interdependencies noted between air quality and precipitation, the performance of models in replicating seasonal spatial patterns and the annual cycle may be related. For example, a model that fails to capture a peak in annual rainfall may also be missing the wet deposition of aerosol caused by the increased precipitation. These links are difficult to determine but are crucial for the continued improvement and development of models.

Performance in simulating airborne particulate matter is very diverse across Coupled Model Intercomparison Project Phase 6 (CMIP6) models (Eyring et al., 2016), as varying representations and parameterisations of differing numbers of aerosol species result in strong differences in AOD between models (Fiedler et al., 2023). For example, Zhao et al. (2022) found that global dust emissions varied by a factor of 5 across 16 CMIP6 models and noted large uncertainties in the simulated dust processes, relating to differences in soil moisture, surface wind speeds, and the dust schemes used. Aerosol processes are represented differently across models, contributing to structural uncertainty. Evaluation of a range of CMIP6 models will aid understanding of historical differences between models and potential future responses.

Over Africa, aerosol emissions are spatially heterogenous, both in terms of the dominant species and magnitude of aerosol loads, as shown in Fig. 1. The frequent biomass burning in tropical forests and intense dust storms from the Sahara Desert are both strong sources of atmospheric pollutants for gaseous and aerosol species (Booyens et al., 2019). There is disagreement in observations and reanalysis datasets for AOD over Africa, for example, the factor of 2 difference in dust burden between the CAMS and MERRA2 reanalyses (Zhao et al., 2022). Issues with representation of atmospheric dust are an important source of intermodel disagreement over Africa, as dust contributes strongly to AOD over northern areas of the continent, as seen in Figure 1. In addition, nitrate aerosol, one of the major contributors to PM$_{2.5}$, as well as AOD, is not modelled by most CMIP6 models (Archer-Nicholls et al., 2023), contributing to strong intermodel spread and worsened performance compared to observations. For example, Pan et al. (2015) found that models without representation of nitrate aerosols severely underestimated AOD over India. Evaluation of PM$_{2.5}$ levels for CMIP6 over Africa has not yet been performed due to a lack of appropriate observational data, and this analysis presents an opportunity to better understand the projections of health risk from PM$_{2.5}$; if the CMIP6 models are producing incorrect PM$_{2.5}$ concentrations, then this could lead to overestimating or underestimating how often dangerous levels of PM$_{2.5}$ are reached, with knock-on effects on estimates of mortality.

CMIP6 models are known to have biases relative to observations in both African rainfall and air quality (Woodward et al., 2022). Precipitation biases occur both spatially, for example, in the southward bias of the tropical rainband, which has persisted for several generations of CMIP (Bock et al., 2020), and temporally, for example, the progression of the monsoons over east and west Africa (Annor et al., 2023; Ayugi et al., 2021). The spatial biases are mostly found over west Africa, whereas biases in the timing of the seasonal cycle are found over both east and west Africa. Finding the root cause of these precipitation biases has proven difficult; previous work has suggested that sea-surface temperature (SST) biases could be to blame (Schwarzwald et al., 2023). Other proposed causes include difficulties simulating the Saharan heat low (SHL) (Dixon et al., 2017), as well as the meridional soil moisture gradient between the Sahara and Gulf of Guinea (Cook, 1999). These previous studies focused on either east or west Africa, while this study applies consistent evaluation approaches across the continent, focussing on both regions.

As there are long-standing biases in precipitation over Africa in CMIP6 that are not explained by coupled SST biases (Schwarzwald et al., 2022), there may also be interlinkages between the performances of precipitation and air quality over Africa, given established interactions between the two. The reasons behind the biases in air quality and precipitation over Africa are not well understood, and this is a clear area for improvement in CMIP6 models. These biases mean that future projections, which also contain high uncertainty for African aerosol emissions under different socioeconomic pathways, are very poorly constrained (Wells et al., 2023). In addition, as the mechanisms through which the two affect each other are not well quantified (Myhre et al., 2013a), the uncertainty in the future evolution of air quality and precipitation is compounded. There is a need for models that can capture air quality and precipitation over Africa accurately to inform policy-making and adaptation strategies in the face of climate change. Due to the risks to human health from changes in air pollution and precipitation, as well as their potential feedback interactions, both factors are discussed jointly in this study.

In this study, our aim is to identify areas of strong and weak performance in CMIP6 models over Africa so that understanding of the biases and overall performance can be used to understand how CMIP6 models respond to future emissions pathways. We demonstrate the performance of CMIP6 models in replicating PM$_{2.5}$ concentrations in 12 cities around Africa through time series presenting both the annual cycle and interannual variability. Expanding into the representation of larger-scale features, we evaluate the performance of CMIP6 models for AOD, examining the seasonal spatial distribution of AOD over the whole of Africa, as well as the performance of regional annual AOD cycles and their interannual variability. Exploring the performance of CMIP6 models over Africa further, the spatial distribution of rainfall by season over the whole of Africa in CMIP6 models is evaluated, as well as the representation of regional African monsoon systems. Changes in PM$_{2.5}$ concentrations are available only from 2016 onwards due to observational data scarcity over Africa (Shindell et al., 2022). This study evaluates all CMIP6 models for which the necessary simulations were available while also highlighting models participating in the Regional Aerosol Model Intercomparison Project (RAMIP) (Wilcox et al., 2023). RAMIP consists of experiments designed to quantify the role of global and regional aerosol emissions changes in near-term projections, including experiments that focus on African emission changes. Given that aerosols have strong impacts on African precipitation, evaluating the performance of these models over Africa will be instrumental in interpreting the results of RAMIP. CMIP6 models are most suitable for large-scale climate studies and are limited in their ability to capture small-scale (e.g. city-scale) processes that impact air quality, which is in the remit of regional models such as the Weather Research and Forecasting model coupled with Chemistry (Grell et al., 2005) or Community Multiscale Air Quality model (US EPA, 2024). However, the use of CMIP6 models provides an opportunity to link air quality to large-scale precipitation and circulation changes, as regional models generally cannot simulate global-scale drivers of air quality changes (Turnock et al., 2020; Guo et al., 2021).

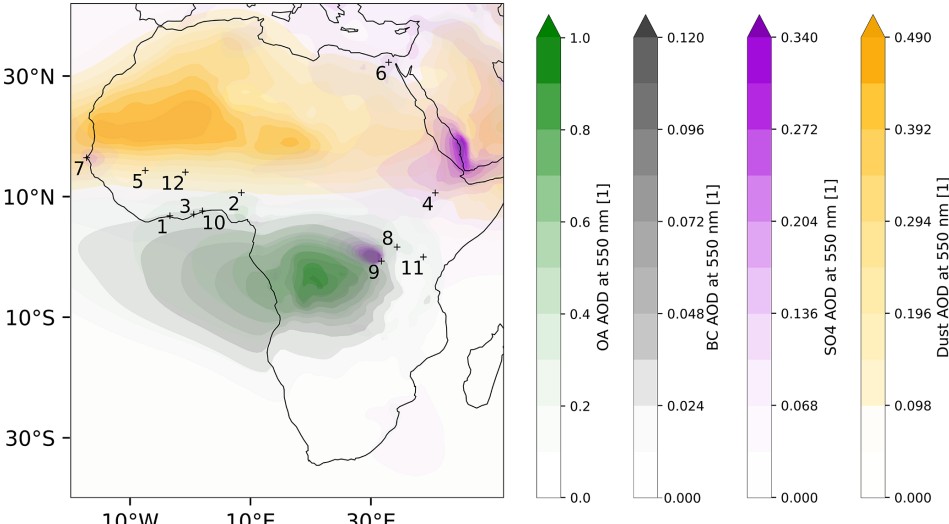

**Figure 1.** AOD contributions from organic aerosol (OA) (green), black carbon (BC) (black), sulfate (SO$_4$) (purple), and dust (yellow) over Africa for SON in the CAMS reanalysis over 1981–2023. Numbered points indicate the location of the PM$_{2.5}$ observation stations used in this analysis (see Table 1 for details of locations). AOD is unitless.

In Sect. 2.1 we introduce the observational and reanalysis datasets used for this study, including the introduction of a new PM$_{2.5}$ observation dataset for Africa. These observations are composed of in situ measurements of PM$_{2.5}$ for several regions in east and west Africa, and, while each location has a different start date for its time series, the earliest measurements are from 2016. This dataset has not yet been compared to CMIP6 model output, and so this study is the first to take the opportunity for observation-based evaluation of CMIP6 performance for PM$_{2.5}$ over Africa. In Sect. 2.2, we introduce the models evaluated. In Sect. 2.3, we discuss the evaluation metrics used. Section 3 shows the performance of the ensembles of models used, arranged by variable. In Sect. 3.2, we evaluate the performance of CMIP6 models against surface observations of PM$_{2.5}$ at different U.S. Embassy locations in Africa to ascertain model biases and intermodel differences. In Sect. 3.3, the AOD performance of CMIP6 models against reanalysis is evaluated, and in Sect. 3.4, the performance of precipitation in CMIP6 models against observations is evaluated. Section 4 summarises these results, discusses notable examples of linkages between biases found, and explores how our results relate to future work on climate responses over Africa.

## 2   Methods

The analysis consists of an evaluation of the present day period in CMIP6, comprising 1981–2023, from a combination of the *historical* and *SSP3-7.0* experiments over Africa, against various observation and reanalysis datasets. The evaluation data are explained in Sect. 2.1, the models are listed in Sect. 2.2, and the methods used are described in Sect. 2.3.

### 2.1   Observational and reanalysis datasets

The datasets used for the evaluation performed in this study are discussed in the following subsections. They include a range of in situ and remote-sensing observations and meteorological and composition reanalyses.

### 2.1.1   Station PM$_{2.5}$ measurements

Data from surface air quality monitors at U.S. Embassy locations in Africa (noted in Table 1) were used for PM$_{2.5}$ evaluation and obtained from the AirNow database (AirNow, 2021). These measurements began in 2016 at the earliest, although some stations record data only from 2023 onwards. There are seven stations in west Africa and four in east Africa, as shown in Fig. 1. The datasets for east Africa are longer, extending back to 2016 and 2017 for Addis Ababa and Kampala. At each air quality monitoring site, Met One Beta Attenuation Monitor 1020 (BAM-1020) sensors are installed, and each sensor is operated by the U.S. State Department. The BAM-1020 is a certified U.S. EPA Federal Equivalent Method monitor for ambient PM$_{2.5}$ concentrations (U. S. Environmental Protection Agency, 2011), which outputs real-time (hourly or finer) measurements of PM$_{2.5}$. The sensor uses beta ray attenuation by a filter-tape medium laden with size-selected particles sampled from ambient air to calculate the PM$_{2.5}$ concentration (Hagler et al., 2022).

### 2.1.2   Reanalysis AOD data

The Copernicus Atmospheric Monitoring Service (CAMS) reanalysis product (for the 2003–2023 time period, with a resolution of 0.75° × 0.75°) is used for AOD evaluation (In-

**Table 1.** Positions of the U.S. Embassies with AirNow PM$_{2.5}$ monitors that are used in this analysis, and the measurement period available for this analysis. The numbers in the final column relate to points shown in Fig. 1 and panel numbers in Fig. 3.

| City | Country | Latitude (° N) | Longitude (° E) | Measurement period | Number |
|------|---------|----------------|-----------------|--------------------|--------|
| Abidjan | Côte d'Ivoire | 5.334 | −3.976 | February 2020–December 2022 | 1 |
| Abuja | Nigeria | 9.041 | 7.477 | February 2020–December 2022 | 2 |
| Accra | Ghana | 5.580 | −0.171 | January 2020–December 2022 | 3 |
| Addis Ababa | Ethiopia | 9.059 | 38.764 | August 2016–August 2022 | 4 |
| Bamako | Mali | 12.630 | −8.019 | October 2019–December 2022 | 5 |
| Cairo | Egypt | 30.041 | 31.234 | May 2022–December 2022 | 6 |
| Dakar | Senegal | 14.745 | −17.526 | February 2022–December 2022 | 7 |
| Kampala | Uganda | 0.300 | 32.592 | February 2017–December 2022 | 8 |
| Kigali | Rwanda | −1.936 | 30.078 | February 2022–December 2022 | 9 |
| Lagos | Nigeria | 6.441 | 3.407 | January 2021–October 2022 | 10 |
| Nairobi | Kenya | −1.234 | 36.811 | July 2021–December 2022 | 11 |
| Ouagadougou | Burkina Faso | 12.305 | −1.497 | January 2022–December 2022 | 12 |

ness et al., 2019). This dataset was used because it offers gridded data, facilitating the evaluation of AOD in regions of Africa with sparse observations from in situ monitoring stations such as AERONET (AERONET, 2024). It assimilates data from multiple satellite retrievals, so AOD in the reanalysis is well constrained by observations. The use of the reanalysis dataset reduces errors due to issues that satellite-based observation datasets such as MODIS face, such as surface brightness over deserts causing a lack of contrast between aerosol signal and the underlying surface brightness (Wagner et al., 2010) or systematic biases being present when clouds interfere with optical measurements (Lee et al., 2013). While the dust optical depth over northern Africa is known to be too low, the CAMS reanalysis has been shown to perform well for mean AOD over this region, as well as capturing climatology and variability (Kapsomenakis et al., 2021). In this study, we evaluate both the AOD due to dust at 550 nm (dust AOD) and the AOD due to aerosols excluding dust at 550 nm (non-dust AOD; i.e. the total AOD minus dust AOD) to identify differences in performance between the dust and non-dust AODs. We evaluate AOD instead of aerosol burden because dust AOD and non-dust AODs are widely available for the majority of CMIP6 models and are directly comparable to the AOD reanalysis, whereas aerosol burden observations are scarce (Fosu-Amankwah et al., 2021). In addition, the AOD in CAMS is more tightly constrained than aerosol burden, through the assimilation of satellite observations (Garrigues et al., 2022), although the speciated AODs are more dependent on the underlying model (Inness et al., 2019).

The evaluation of dust AOD should be understood in the context that dust AOD is not directly constrained by observations. Dust AOD in CAMS is a derived field, inferred through aerosol speciation methods that use measurements of the fine-mode fraction, Ångström exponent, and single-scattering albedo. Errors in reanalysis dust AOD can be introduced through assumptions about aerosol speciation, the assumptions and errors during the measurement itself, and

model-dependent processes with their associated uncertainties (Xian et al., 2020; Zhao et al., 2022).

Zhao et al. (2022) notes that accurate representation of dust AOD in reanalysis relies on simulating correct amounts of dust relative to other aerosol species. Dust AOD is better represented in areas where dust dominates, for example, over the Sahara, but is less well represented in regions where there are similar contributions from different aerosol species (Zhao et al., 2022).

Uncertainties in dust AOD for different datasets are discussed in detail in Vogel et al. (2022), which notes regional uncertainty for dust AOD in CAMS and MERRA-2. Speciated AOD is also subject to uncertainties in total AOD, and north Africa is one of the regions of highest AOD uncertainty for the satellite and reanalysis datasets (Vogel et al., 2022). In addition, reanalysis and observational datasets show worse agreement over bright and heterogeneous surfaces such as northern Africa (Garrigues et al., 2022).

### 2.1.3 Precipitation observation and reanalysis datasets

The Climate Hazards group InfraRed Precipitation with Station data (CHIRPS) (Funk et al., 2015) dataset, a land-only gauge-based rainfall dataset produced from blended station data (1981–2023, 0.05° × 0.05°), CRU (Mitchell and Jones, 2005) (1901–2017, 0.5° × 0.5°), ERA5 (Hersbach et al., 2020) (1940–2023, 0.25° × 0.25°), and GPCP (Huffman et al., 2023) (1979–2023, 2.5° × 2.5°), was considered when choosing a reference observational or reanalysis dataset for this analysis. The CHIRPS product was chosen because it was specifically designed to monitor rainfall in the tropics, and it has been shown that CHIRPS has good performance relative to other datasets used to evaluate and monitor Africa and the tropics (Ayehu et al., 2018). Beck et al. (2017) found that CHIRPS ranked among the best performers in capturing precipitation indices and long-term precipitation means but also noted that it underestimated the peak

magnitude of rainfall and produced spurious drizzle – which was found to be a common bias among the reanalyses evaluated. CHIRPS is intended for environmental monitoring, so its performance in capturing low-frequency climate variability is not well known. However, trends are seen to be well represented over east Africa (Peterson et al., 2014), and while CHIRPS overestimates the frequency of rainfall, the rainfall contributed by these extra events is small (Diem et al., 2019).

## 2.2 Model datasets

In this study, we make use of all CMIP6 models for which the relevant outputs are available for the *historical* and *SSP3-7.0* simulation to cover the 1981–2023 period. This means that there are more models evaluated for precipitation than for AOD and more evaluated for AOD than for PM$_{2.5}$ concentrations. However, understanding the range of model performance across the CMIP6 ensemble, without restricting the study only to models with all aerosol variables available, was a priority for the choice of models. We use the *historical* experiment from CMIP6, which includes time-varying emissions of greenhouse gases, aerosols, and ozone, along with volcanic and solar forcing for the period 1981–2014, and the "regional rivalry" shared socioeconomic pathway, *SSP3-7.0*, experiment from ScenarioMIP (O'Neill et al., 2016) for the period 2015–2023. The choice of a single scenario, *SSP3-7.0*, does not bias the results here, as aerosol emissions do not diverge in the period used (Gidden et al., 2018). These datasets were concatenated before the analysis. The models used in the study, along with their nominal atmospheric and ocean resolutions, are shown in Table 2. The numbers assigned to each model are used to label them in later figures. In addition to the general evaluation of CMIP6 models, we highlight the performance of models participating in the Regional Aerosol Model Intercomparison Project (RAMIP). RAMIP includes experiments to explore the climate and air quality responses to near-future changes in African emissions of sulfur dioxide, black carbon, and organic carbon. As the African climate has already been shown to be sensitive to remote aerosol changes (Scannell et al., 2019; Dong et al., 2014), it is anticipated that the African response to both local and remote changes in aerosol emissions will be a key focus of the analysis of RAMIP experiments. Evaluating the performance of participating models in replicating regional climate and air quality over Africa is therefore important before further work.

## 2.3 Methodology

The majority of the analysis in this study uses monthly mean outputs; while daily mean outputs are better able to capture sub-seasonal variability, these were not available for most models for variables such as speciated AOD. However, daily mean outputs are used for the evaluation of daily precipitation behaviour in order to investigate the representation of wet and dry days as well as rainfall intensity in CMIP6 models.

For the observational and reanalysis datasets used, the maximum available time domain over the 1981–2023 period is used to minimise the influence of low-frequency variability and better capture the characteristics of the annual cycles. The available time domain is different for each dataset, as discussed below. For AOD, because part of the available reanalysis dataset is past the end date of the *historical* CMIP runs, ScenarioMIP (O'Neill et al., 2016) experiments are used, following the *SSP3-7.0* scenario. Over the time span that this covers (2015 to 2023), the differences in emissions inventories between SSPs cause minimal differences in AOD at the continental scale for Africa (Lund et al., 2019).

The large bounding box used for this study (40° S, 40° N, 20° W, 50° E) is based on the subregion boxes defined by the IPCC (Chen et al., 2021), covering the whole of the African continent. Areas of particular focus are the east and west African monsoon regions; both of these regions feature a complex climatology with known biases and strong future climate responses. In this study, west Africa is defined by 10° S–15° N, 20° W–25° E, and east Africa by 5° S–15° N, 27–46° E, as these capture the key features of the monsoon climatology.

Monthly means are used for evaluation of the annual cycle for each model. In addition, the spatial distribution of rainfall is evaluated seasonally, focusing on June, July, and August (JJA) and December, January, and February (DJF) to capture the periods of maximum northward and southward displacement of the intertropical convergence zone (ITCZ). For more specific regional analysis, it would be preferable to instead use the relevant monsoon periods; for example, over west Africa, this would be July, August, and September (JAS), as seen in Fig. 12. However, DJF and JJA were chosen for studying the performance over a larger region while still demonstrating the seasonal shifts in spatial distributions.

As discussed in Sect. 2.1.3, the CHIRPS precipitation observational dataset is available only over land. Therefore, for calculating the bias and pattern correlation associated with each model, a land mask corresponding to areas where the land mass fraction for that model was greater than 0.5 was applied.

The retrieval of PM$_{2.5}$ data was more complex; of the models for which PM$_{2.5}$ data were available, EC-Earth3-AerChem, GFDL-ESM4, GISS-E2-1-G, IPSL-CM5A2-INCA, MIROC6, and NorESM2-LM output PM$_{2.5}$ directly, while CESM2 and UKESM1-0-LL output mass mixing ratios (MMRs) of seven PM$_{2.5}$ components, including sulfate (SO$_4$), organic carbon (OA), black carbon (BC), sea salt (SS), dust (DU), nitrate, and ammonium. The mixing ratios of PM$_{2.5}$ and its components were obtained using MMR diagnostics for the *SSP3-7.0* experiments from 2015 to 2023.

Observed measurements of PM$_{2.5}$ in the AirNow database in Africa began in August 2016. In cases where the CMIP6 models do not provide PM$_{2.5}$ directly, we calculate it

**Table 2.** List of CMIP6 models used. Numbers attributed to each model are used to label each model in later figures. RAMIP models are indicated using ***bold italics***.

| Centre | Model | Number | Data reference | Model reference | Number of ensemble members | Nominal atmospheric resolution | Nominal ocean resolution |
|---|---|---|---|---|---|---|---|
| CSIRO-ARCCSS | ACCESS-CM2 | 1 | Dix et al. (2019) | Dix et al. (2019) | 10 | 250km | 100km |
| CSIRO | ACCESS-ESM1-5 | 2 | Ziehn et al. (2019) | Ziehn et al. (2020) | 40 | 250km | 100km |
| AWI-CM | AWI-CM-1-1-MR | 3 | Semmler et al. (2018) | Semmler et al. (2020) | 5 | 100km | 25km |
| AWI | AWI-ESM-1-1-LR | 4 | Danek et al. (2020) | Sidorenko et al. (2015) | 1 | 250km | 50km |
| BCC | BCC-CSM2-MR | 5 | Wu et al. (2018) | Wu et al. (2019) | 3 | 100km | 50km |
| BCC | BCC-ESM1 | 6 | Zhang et al. (2018) | Wu et al. (2020) | 3 | 250km | 50km |
| CAMS | CAMS-CSM1-0 | 7 | Rong (2019) | Rong et al. (2018) | 3 | 100km | 100km |
| CCCma | CanESM5 | 8 | Swart et al. (2019a) | Swart et al. (2019d) | 40 | 500km | 100km |
| ***CCCma*** | ***CanESM5-1*** | 9 | Swart et al. (2019b) | Swart et al. (2019b) | 20 | 500km | 100km |
| CCCma | CanESM5-CanOE | 10 | Swart et al. (2019c) | Christian et al. (2022) | 3 | 500km | 100km |
| CAS | CAS-ESM2-0 | 11 | Zhang et al. (2020) | Zhang et al. (2020) | 3 | 100km | 100km |
| NCAR | CESM2-FV2 | 12 | Danabasoglu (2019a) | Danabasoglu et al. (2020) | 3 | 100km | 100km |
| ***NCAR*** | ***CESM2*** | 13 | Danabasoglu (2019c) | Danabasoglu et al. (2020) | 11 | 250km | 100km |
| NCAR | CESM2-WACCM | 14 | Danabasoglu (2019b) | Danabasoglu et al. (2020) | 3 | 100km | 100km |
| THU | CIESM | 15 | Huang (2019) | Lin et al. (2020) | 3 | 100km | 50km |
| CMCC | CMCC-CM2-HR4 | 16 | Scoccimarro et al. (2020) | Cherchi et al. (2019) | 1 | 100km | 25km |
| CMCC | CMCC-CM2-SR5 | 17 | Lovato and Peano (2020) | Cherchi et al. (2019) | 1 | 100km | 100km |
| CMCC | CMCC-ESM2 | 18 | Peano et al. (2020) | Peano et al. (2020) | 1 | 100km | 100km |
| CNRM-CERFACS | CNRM-CM6-1 | 19 | Voldoire (2018) | Voldoire et al. (2019) | 30 | 250km | 100km |
| CNRM-CERFACS | CNRM-CM6-1-HR | 20 | Voldoire (2019) | Voldoire et al. (2019) | 1 | 100km | 25km |
| ***CNRM-CERFACS*** | ***CNRM-ESM2-1*** | 21 | Seferian (2018) | Séférian et al. (2019) | 8 | 250km | 100km |
| E3SM-Project | E3SM-1-0 | 22 | Bader et al. (2019a) | Golaz et al. (2019) | 1 | 100km | 50km |
| E3SM-Project | E3SM-1-1 | 23 | Bader et al. (2019b) | Burrows et al. (2020) | 1 | 100km | 50km |
| E3SM-Project | E3SM-1-1-ECA | 24 | Bader et al. (2020) | Burrows et al. (2020) | 1 | 100km | 50km |
| E3SM-Project | E3SM-2-0 | 25 | Bader et al. (2023) | Golaz et al. (2022) | 1 | 100km | 50km |
| EC-Earth Consortium | EC-Earth3 | 26 | EC-Earth Consortium (EC-Earth) (2019b) | Döscher et al. (2021) | 74 | 100km | 100km |
| ***EC-Earth Consortium*** | ***EC-Earth3-AerChem*** | 27 | EC-Earth Consortium (EC-Earth) (2020) | van Noije et al. (2021) | 4 | 100km | 100km |
| EC-Earth Consortium | EC-Earth3-CC | 28 | EC-Earth Consortium (EC-Earth) (2021) | Döscher et al. (2022) | 1 | 100km | 100km |
| EC-Earth Consortium | EC-Earth3-Veg | 29 | EC-Earth Consortium (EC-Earth) (2019a) | Döscher et al. (2022) | 12 | 100km | 100km |
| CAS | FGOALS-f3-L | 30 | YU (2019) | He et al. (2020) | 3 | 100km | 100km |
| CAS | FGOALS-g3 | 31 | Li (2019) | Li et al. (2020) | 3 | 250km | 100km |
| ***NOAA-GFDL*** | ***GFDL-CM4*** | 32 | Guo et al. (2018) | Held et al. (2019) | 1 | 100km | 25km |
| NOAA-GFDL | GFDL-ESM4 | 33 | Krasting et al. (2018) | Dunne et al. (2020) | 3 | 100km | 50km |
| ***NASA-GISS*** | ***GISS-E2-1-G (p3)*** | 34 | GISS (2018) | Miller et al. (2021) | 10 | 250km | 100km |
| NASA-GISS | GISS-E2-1-G-CC | 35 | GISS (2019a) | Kelley et al. (2020), Miller et al. (2021) | 1 | 250km | 100km |
| NASA-GISS | GISS-E2-1-H (p3) | 36 | GISS (2019b) | Miller et al. (2021) | 1 | 250km | 100km |
| NASA-GISS | GISS-E2-2-G | 37 | NASA GISS (2019a) | Rind et al. (2020) | 1 | 250km | 100km |
| NASA-GISS | GISS-E2-2-H | 38 | NASA GISS (2019b) | Rind et al. (2020) | 1 | 250km | 100km |
| MOHC | HadGEM3-GC31-LL | 39 | Ridley et al. (2019a) | Andrews et al. (2020), Kuhlbrodt et al. (2018) | 55 | 250km | 100km |
| MOHC | HadGEM3-GC31-MM | 40 | Ridley et al. (2019b) | Andrews et al. (2020) | 4 | 100km | 25km |
| CCCR-IITM | IITM-ESM | 41 | Narayanasetti et al. (2019) | Krishnan et al. (2021) | 1 | 250km | 100km |
| INM | INM-CM4-8 | 42 | Volodin et al. (2019a) | Volodin (2010) | 1 | 100km | 100km |
| INM | INM-CM5-0 | 43 | Volodin et al. (2019b) | Volodin et al. (2017), Volodin and Kostrykin (2016) | 10 | 100km | 50km |
| IPSL | IPSL-CM5A2-INCA | 44 | Sepulchre et al. (2020) | Boucher et al. (2020a) | 1 | 500km | 250km |
| IPSL | IPSL-CM6A-LR | 45 | Boucher et al. (2021b) | Boucher et al. (2020b) | 33 | 250km | 100km |
| IPSL | IPSL-CM6A-LR-INCA | 46 | Boucher et al. (2021a) | Boucher et al. (2020b) | 1 | 250km | 100km |
| NIMS-KIMA | KACE-1-0-G | 47 | Lee et al. (2020) | Byun et al. (2019) | 2 | 250km | 100km |
| KIOST | KIOST-ESM | 48 | Pak et al. (2021) | Kim et al. (2019) | 1 | 250km | 100km |
| UA | MCM-UA-1-0 | 49 | Stouffer (2019) | Stouffer (2019) | 1 | 250km | 250km |
| MIROC | MIROC-ES2H | 50 | Watanabe et al. (2021) | Kawamiya et al. (2020) | 30 | 250km | 100km |
| ***MIROC*** | ***MIROC6*** | 51 | Tatebe and Watanabe (2018) | Tatebe et al. (2019) | 30 | 250km | 100km |
| MIROC | MIROC-ES2L | 52 | Hajima et al. (2019) | Hajima et al. (2020) | 50 | 500km | 100km |
| HAMMOZ-Consortium | MPI-ESM-1-2-HAM | 53 | Neubauer et al. (2019) | Neubauer et al. (2019) | 3 | 250km | 250km |
| ***MRI*** | ***MRI-ESM2-0*** | 54 | Yukimoto et al. (2019b) | Yukimoto et al. (2019a) | 3 | 100km | 100km |
| NCC | NorCPM1 | 55 | Bethke et al. (2019) | Bethke et al. (2019) | 30 | 250km | 100km |
| ***NCC*** | ***NorESM2-LM*** | 56 | Seland et al. (2019) | Seland et al. (2020) | 3 | 250km | 100km |
| NCC | NorESM2-MM | 57 | Bentsen et al. (2019) | Seland et al. (2020) | 3 | 100km | 100km |
| AS-RCEC | TaiESM1 | 58 | Lee and Liang (2020) | Wang et al. (2021) | 2 | 100km | 25km |
| ***MOHC*** | ***UKESM1-0-LL*** | 59 | Tang et al. (2019) | Sellar et al. (2019) | 16 | 250km | 100km |
| MOHC | UKESM1-1-LL | 60 | Walton et al. (2022) | Mulcahy et al. (2023) | 1 | 250km | 100km |

from speciated mass mixing ratios at the surface following Turnock et al. (2020):

$$PM_{2.5} = BC + SO_4 + OA + (0.25 \times SS) + (0.1 \times DU), \quad (1)$$

where BC, SO$_4$, OA, SS, and DU are the surface mass mixing ratios of black carbon, sulfate, organic aerosol, sea salt, and dust, respectively.

This method of calculating PM$_{2.5}$ levels relies on simplified assumptions about size distributions of aerosol species – for example, that all BC particles have sub-2.5 μm diameters, whereas only 10 % of dust particles do. However, the size distribution of the species will vary between each model, so this assumption will not always be valid and therefore can introduce errors. However, these values for the contribution of each aerosol species to PM$_{2.5}$ have been shown to be effective for most regions in Turnock et al. (2020), except for ocean regions where assumptions about SS size distributions break down. In addition to errors introduced by the calculation itself through size distribution assumptions, model biases in the MMRs of aerosol species will be reflected in biases in the calculated PM$_{2.5}$ values.

Regions with strong dust emissions, such as the Sahara, are associated with particularly high variability in PM$_{2.5}$ across models, when evaluating using the calculated PM$_{2.5}$ (Turnock et al., 2020). Outside of these high dust emission regions, Turnock et al. (2020) showed that the PM$_{2.5}$ levels calculated show higher model agreement, though this finding excludes Africa, which was not included in that portion of the analysis due to limited observations. The largest source of model diversity in calculated PM$_{2.5}$ for the majority of stations over Africa is differences in dust, and sea salt also becomes a more important source of model diversity in coastal locations (Turnock et al., 2020). PM$_{2.5}$ concentrations tend to be underestimated across most regions, which Turnock et al. (2020) notes may be due to the exclusion of nitrate aerosols from the calculation formula, as well as biases in other aerosol sources and processes.

We computed monthly mean simulated PM$_{2.5}$ from the CMIP6 models and corresponding monthly ensemble mean surface PM$_{2.5}$ observations at each location, after all models were regridded to a common grid (1° × 1° resolution), before performing PM$_{2.5}$ calculations using the nearest available grid point. The performance of the models was evaluated using the coefficient of determination ($R^2$), root mean squared error (RMSE), and mean absolute error (MAE). All regridded data are interpolated to a 1° × 1°-resolution grid unless otherwise specified.

For precipitation evaluation, after time-averaging the datasets to 2D fields, pattern correlation was performed. This was done by regridding the model and observational fields to a 1° × 1° grid, then connecting the rows of data to make 1D datasets. The Pearson correlation coefficient was calculated from these. The same method was used to calculate the pattern correlation coefficients for the AOD, correlating against the reanalysis dataset. This calculation shows how well the pattern of modelled data matches that of the observation or reanalysis over the domain. For precipitation, this is useful because we are interested in the representation of rainfall location and progression, as this may influence aerosol–precipitation interactions and the precipitation response to regional climate forcing. Pattern correlation is also useful for AOD, as it demonstrates how well the locations and sizes of areas of high AOD are captured. However, the pattern correlation provides no indication of whether the average magnitude of rainfall or AOD is correct. Therefore, to indicate the performance of the magnitude of the fields, the RMSE is also shown for the spatial plots.

## 3 Results

In this section, we evaluate the performance of CMIP6 models in replicating PM$_{2.5}$, AOD, and precipitation compared to observations and reanalyses. We first start with a summary figure (Sect. 3.1) and then provide details for PM$_{2.5}$, AOD, and precipitation performances.

### 3.1 Summary of results

An overview of the results for the evaluation of precipitation and AOD performance can be found in Fig. 2. This figure provides an overview of each model's performance in non-dust and dust AOD seasonal spatial patterns, AOD climatology over east and west Africa, precipitation seasonal spatial pattern, monsoon progressions in east and west Africa (demonstrated through zonal mean precipitation climatology), and daily precipitation distributions over east and west Africa. Models that perform consistently well in seasonal spatial AOD patterns are MRI-ESM2-0 and IPSL-CM6A-LR, and these models also perform well in seasonal spatial precipitation patterns. In contrast, CESM2, CESM2-WACCM, and NorESM2-MM all perform consistently well for their precipitation patterns but have difficulties replicating AOD patterns, particularly for dust AOD when compared to the performance of other models. These results are discussed in more detail in the relevant sections.

### 3.2 PM$_{2.5}$

For evaluating the behaviour of PM$_{2.5}$ in the models, the behaviour of the PM$_{2.5}$ time series for 12 different cities is examined. Figure 3 shows the comparison of time series of PM$_{2.5}$ in the CMIP6 models and surface observations from reference monitors at 12 U.S. Embassy locations in Africa. In this figure, the interannual variations are not expected to match with those of observations, as individual years in CMIP are not designed to correspond to the observed years. The CMIP6 models capture the seasonal cycle better in west Africa (1, 2, 3, 5, 7, 10, 12) than in east Africa (4, 8, 9, 11). This is also evident in the $R^2$, RMSE, and MAE values shown in Table S1 in the Supplement. West Africa's PM$_{2.5}$

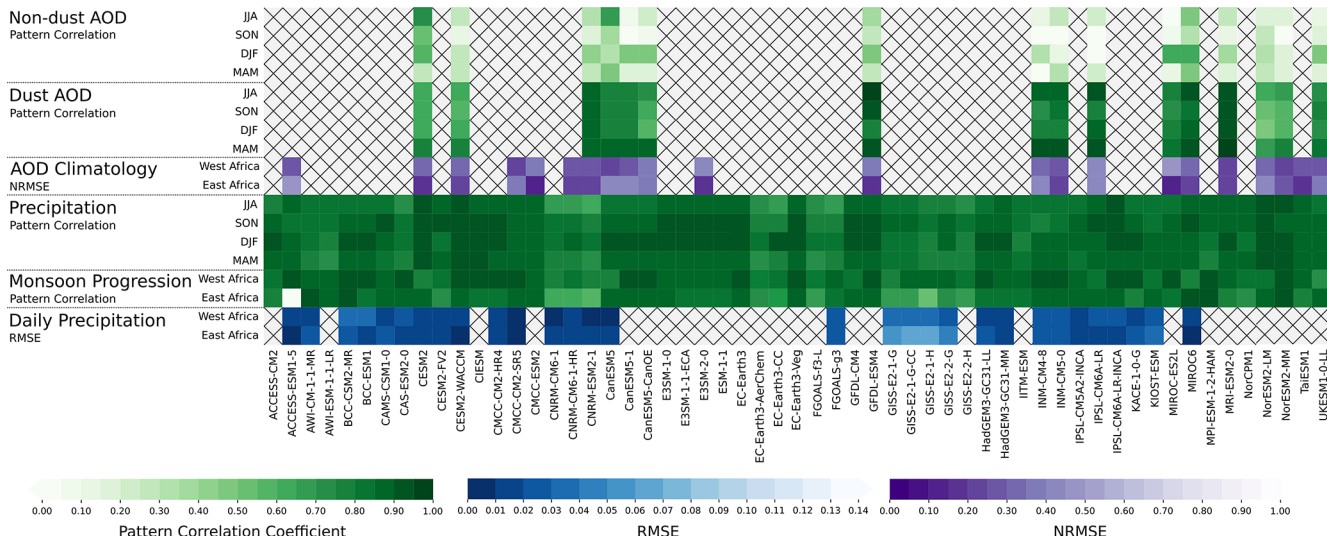

**Figure 2.** Performance indicators across precipitation and AOD for all models evaluated in this study. Green indicates the pattern correlation coefficient, blue indicates the RMSE, and purple indicates the normalised root mean square error (NRMSE), with more saturated colours indicating better performance for all colour scales.

levels are largely influenced by the Harmattan season, which is associated with increased dust and a northeasterly flow over much of north Africa, carrying dust from the Sahara Desert (Anuforom, 2007). Therefore, accurate PM$_{2.5}$ simulation in a model is closely tied to its representation of atmospheric circulation and magnitude of dust emission, the latter of which differs strongly between models (Zhao et al., 2022). Conversely, east Africa's PM$_{2.5}$ concentrations are more heavily influenced by local emissions (Kalisa et al., 2023), and thus model climatologies are dependent on the accuracy of emissions inventories.

From Fig. 3, it can be seen that the models demonstrate a strong diversity in magnitude for PM$_{2.5}$ concentrations. It could be expected that the modelled PM$_{2.5}$ would be too low compared to that of the observations, as the observation stations are in urban areas that are not modelled by the CMIP6 models and, as Schutgens et al. (2016) demonstrates, increasing grid box sizes and distances from observation stations to grid points result in increased errors. There is a mixture of both positive and negative PM$_{2.5}$ concentration biases.

CMIP6 models can be seen to generally capture the PM$_{2.5}$ annual cycle well. Cairo is a notable exception, as models underestimate PM$_{2.5}$ relative to the surface observations and do not match the observed seasonal cycle. However, there is a very small time period for which PM$_{2.5}$ observed data are available over Cairo ($<$ 1 year), and future observational data may show stronger agreement with the models. IPSL-CM5A2-INCA overestimates surface PM$_{2.5}$ observations in all locations, especially during periods of increased PM$_{2.5}$ levels. This may relate to biases in daily precipitation behaviour, as IPSL-CM5A2-INCA underestimates the upper end of daily rainfall amounts in both east and west

Africa – daily precipitation behaviour is discussed further in Sect. 3.4.3. However, MIROC6 displays similar behaviour in daily precipitation to IPSL-CM5A2-INCA and yet has the opposite bias; MIROC6 underestimates PM$_{2.5}$ at most observation stations. However, MIROC6, despite underestimating the upper end of daily rainfall, exhibits wet biases for most seasons over much of Africa, especially west Africa, where biases in IPSL-CM5A2-INCA for the same region are less clear.

Most models perform best in Accra, Ouagadougou, Abidjan, Abuja, and Lome, with $R^2$ ranging from 0.35 to 0.96 and the mean absolute error (MAE) ranging from 2.42 to 20.07 µg m$^{-3}$. All the models perform poorly in Kampala, Nairobi, Kigali, Addis Ababa, Bamako, and Dakar, with the largest RMSE and MAE in Bamako, ranging from 26.91 to 31.44 µg m$^{-3}$ and 22.01 to 25.07 µg m$^{-3}$, respectively. It is notable that the cities with poor performance are generally located in east Africa, with the exception of Bamako and Dakar. This is in agreement with results for the larger-scale AOD analysis shown in Sect. 3.3, where representation of the annual cycle of AOD is found to be generally poorer over east Africa compared to west Africa.

The complex monsoon climatology impacts the levels of PM$_{2.5}$, so performance in simulating the east African monsoon (EAM) and west African monsoon (WAM), as discussed in Sect. 3.4.2 and 3.4.1, will impact wet deposition rates and thus may be related to model performance in replicating PM$_{2.5}$ concentrations. As the east African monsoon is found to be modelled less accurately than the west African monsoon, this may explain why the PM$_{2.5}$ annual cycle is not captured as well over east African cities, for example, Kigali and Kampala, compared to west African cities, for example,

Bamako and Abuja. In addition, precipitation will affect soil moisture, which will change dust emissions and thus alter PM$_{2.5}$ levels. This is more likely to be related to PM$_{2.5}$ levels over west Africa, where dust emissions play a dominant role in PM$_{2.5}$ levels.

As daily precipitation relates to air quality (Wang et al., 2023), a comparison of observed rainfall and air quality datasets in the future may lead to further understanding of these interactions when more air quality data become available over Africa. Initial comparison of the datasets showed the expected relationship of increased precipitation coinciding with decreased PM$_{2.5}$ in west Africa, though the relationship is less clear in east Africa. In some cities, such as Addis Ababa and Cairo, increased precipitation is seen to coincide with increased PM$_{2.5}$, which is opposite to the expected relationship; the reason for this correlation has not been determined. The comparison of the datasets is shown in the supplementary information.

## 3.3  Aerosol optical depth

For evaluating the behaviour of AOD in the models, we consider the accuracy of both dust and non-dust AOD distributions separately, as they have different contributions of uncertainty and separate analysis makes the origins of biases easier to identify. Some models output speciated non-dust aerosols, such as black carbon and sulfate, but the majority do not, so this analysis is split into dust and non-dust contributions only. We also examine the accuracy of the seasonal cycle of AOD over subregions in Africa.

The key results of AOD evaluation can be seen in Figs. 4 and 5. They show the reanalysis (CAMS) distribution of dust and non-dust AOD during September, October, and November (SON), the multi-model mean (MMM), the intermodel standard deviation, the bias in the MMM, and the biases in the models with the least and most deviation from the spatial pattern of dust and non-dust AOD in CAMS. SON is used as the example season for the AOD analysis, as it shows the strongest intermodel diversity and lowest overall performance for both the dust and non-dust AODs.

Figure 4 shows that the MMM, while generally in agreement with reanalysis, does not fully capture the distribution of dust AOD during SON. The northward and westward extent of the high dust AOD over the Sahara is not well captured, with a positive bias over northeast Africa and a negative bias over the northwest. Strong intermodel variability over the central Saharan region may be due to the different interactive dust schemes used by many CMIP6 generation models, causing strong differences in the magnitude of dust aerosol. There is also some intermodel disagreement in wind speed over the region; because interactive dust emissions rely on surface wind speeds, this will contribute to differences in interactive dust emissions. Intermodel differences in other meteorological conditions that are not shown here, such as soil moisture, will also contribute to the diver-

sity seen (Zhao et al., 2022). However, the initial analysis of surface wind speed, which was found to be the dominant driver of dust emissions for a large group of CMIP6 models in Zhao et al. (2022), showed that outliers in surface wind speed behaviour (for example, INM-CM5-0 and UKESM1-0-LL) did not correspond to outliers in dust AOD (for example, the CESM2 family and IPSL-CM6A-LR). Though further analysis is necessary, this may indicate that differences in dust AOD arise mainly from the different dust schemes between models rather than differences in the meteorological conditions for each model. The analysis noted is shown in the supplementary information.

The closest model to the reanalysis, GFDL-ESM4, captures the dust AOD well. While the biases that are present still show a negative (positive) dust AOD bias over northwest (northeast) Africa, the model performs well overall for SON, with a pattern correlation of 0.95 and an RMSE of 0.04.

The model furthest from the reanalysis, NorESM2-LM, has a positive bias in dust AOD of $> 0.8$ over northeast Africa extending into central Africa and a strong negative bias over northwest Africa, as well as a positive AOD bias over southern Africa. The latitudinal band of maximum AOD also shows a southward bias. NorESM2-LM is in the CESM2 family of CMIP6 models (CESM2, CESM2-FV2, NorESM2-MM, etc.), which tend to produce the correct amount of dust but confine it to only a small number of grid points (Zhao et al., 2022). This causes a common bias in these models, where the dust AOD over those areas is too high and the AOD elsewhere in dust-affected regions is too low, which is found in this analysis in the area of positive dust AOD bias over northwest Africa. Some research has noted that this bias may relate to wind speed or soil moisture (Zhao et al., 2022), though co-located areas of strong biases in dust AOD and these variables were not identified in this analysis. Overall, this model struggles to replicate the pattern of dust AOD over Africa during SON, which is reflected in its lower pattern correlation coefficient of 0.52 and RMSE of 0.15.

The dust AOD spatial distribution over Africa in SON is well captured by most climate models. The strongest biases relate to the area of high dust AOD over the Sahara being too far eastward and the magnitude of peak AOD being overestimated, which is almost entirely contributed to by CESM2 family models, with their positive AOD bias noted above. The differences shown in AOD patterns contribute to uncertainty in effective aerosol radiative forcing (Kalisoras et al., 2024; Thornhill et al., 2021; Zelinka et al., 2023) and thus add uncertainty to climate responses over Africa. In addition, as aerosol forcing relates to changes in regional precipitation patterns and monsoon dynamics (Shonk et al., 2020; Shindell et al., 2012; Williams et al., 2022), differences in AOD patterns are strongly relevant to projected climate responses.

For non-dust AOD, Fig. 5 shows that the MMM does not capture the distribution of non-dust AOD during SON well. The MMM and MMM bias in Fig. 5b and c show a positive AOD bias over the west coast of west Africa, which may

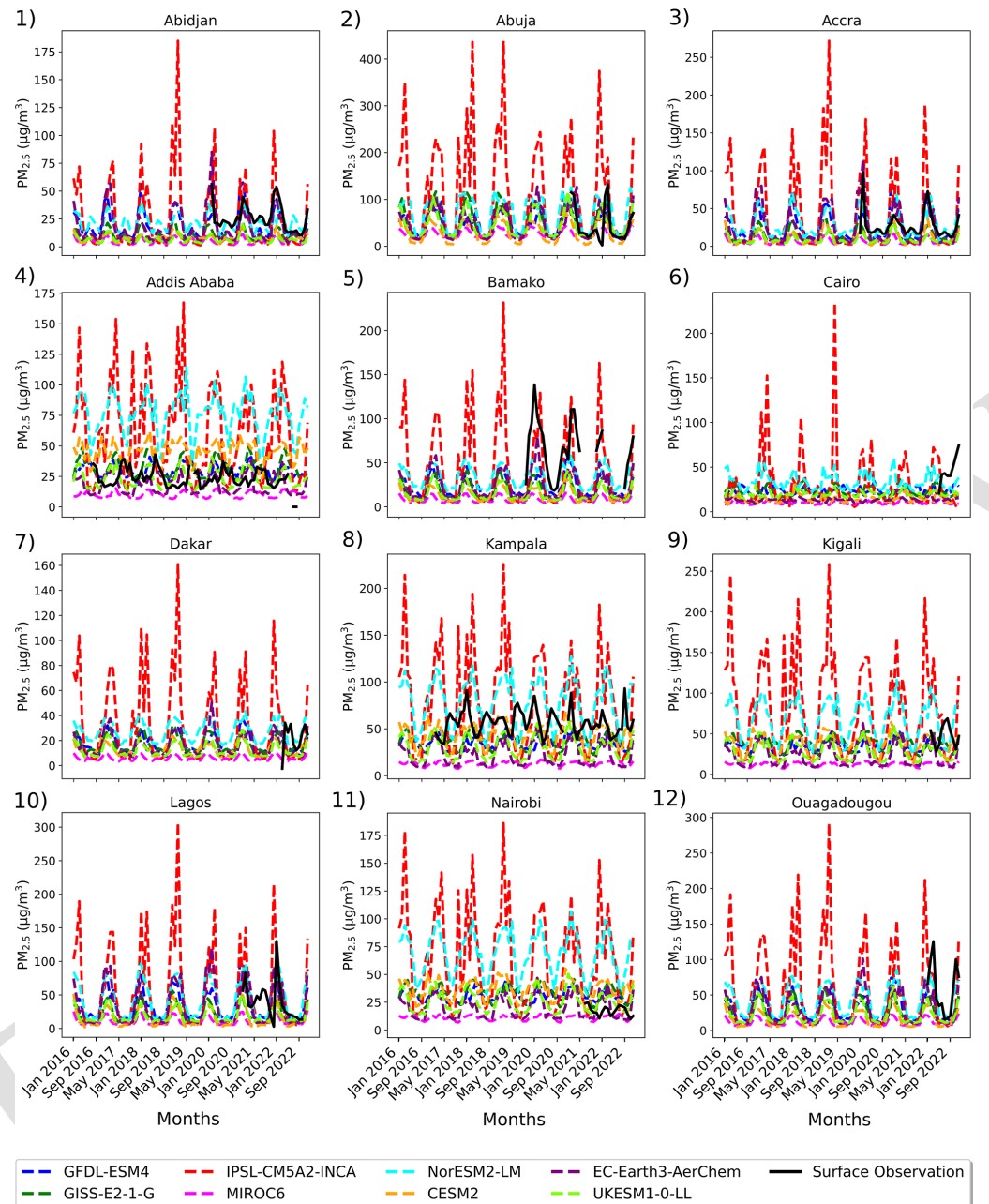

**Figure 3.** Comparison of CMIP6 models (dashed coloured lines) and surface PM$_{2.5}$ observations (solid black lines) with reference monitors at U.S. Embassy locations in Africa. Note the changing $y$ axis between plots. A common $x$ axis is used, though not all of the dates are covered for each location by the available observational dataset. Numbers refer to the station locations mapped in Fig. 1 and detailed in Table 1. Units are µg m$^{-3}$.

be due to coincident anomalous westerly winds. As these act to weaken the southeasterly flow from over the Gulf of Guinea, less clean air is advected, which may lead to the positive AOD bias. The area of high AOD over southern central Africa, associated with biomass burning, is missing from the MMM.

The closest model to the reanalysis, CESM2, captures some of the non-dust AOD spatial pattern. The negative AOD

bias found in the MMM over southern central Africa is still present, though biases elsewhere are reduced. It is notable that the CESM2 model family that performed poorly for the dust AOD pattern performs best in the non-dust AOD pattern evaluation, though none of the models analysed produce non-dust AOD similar to that of the reanalysis. CESM2 is found to have a pattern correlation of 0.50 and an RMSE of 0.08.

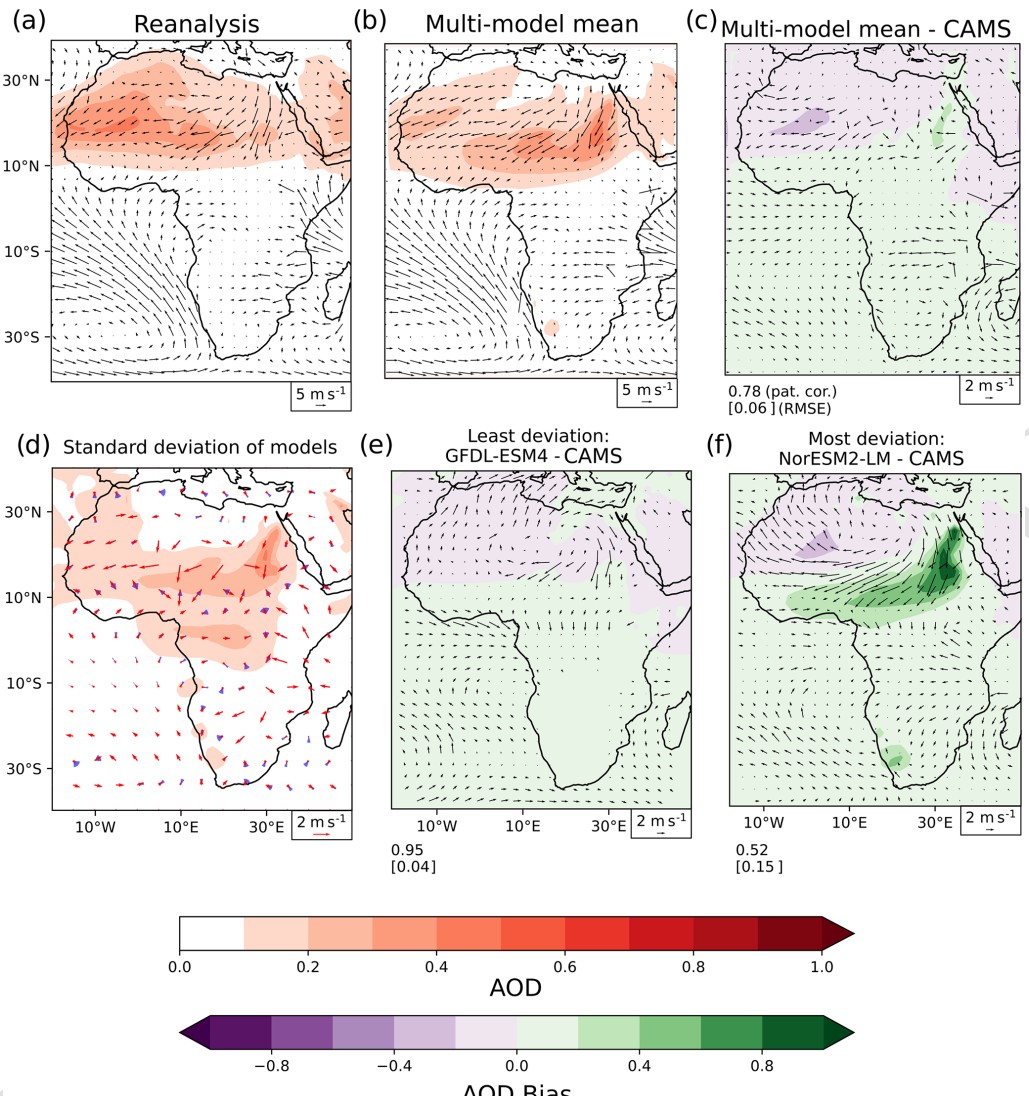

**Figure 4.** SON mean dust AOD and lower-tropospheric (925 hPa) winds for 2003–2023 in **(a)** observations (CAMS/ERA5) and **(b)** CMIP6 MMM. **(c)** CMIP6 MMM bias against observations and **(d)** intermodel standard deviation for dust AOD (shading) and wind speed along the mean wind direction (red) and wind direction (blue). Mean dust AOD and wind fields in the models with the **(e)** least (GFDL-ESM4) and **(f)** most (NorESM2-LM) deviation from CAMS as determined by pattern correlation. Pattern correlation and RMSE compared to CAMS are shown below panels **(c)**, **(e)**, and **(f)**. Note that the reference vectors for the 925 hPa winds differ between panels. Winds in panel **(d)** have been regridded to a 4° × 4° grid for clarity.

For the model furthest from the reanalysis, CanESM5-1, the negative AOD bias found in the MMM over southern central Africa is present. In addition, a strong positive (> 0.8) AOD bias is found over the UAE – this AOD bias covers a large area in the northeast and is not limited to Africa. CanESM5-1 was found to have a negligible pattern correlation coefficient of −0.08 and an RMSE of 0.25.

The non-dust AOD spatial distribution over Africa in SON is not well captured by the climate models analysed here. The strongest bias relates to a negative AOD anomaly found over southern central Africa.

Figure 6 shows the comparative performance of the models when ranked by pattern correlation compared to CAMS for each season, with the RAMIP models highlighted in red, for dust AOD and non-dust AOD. The mean performance of the CMIP6 models is lowest in DJF for the dust AOD and in SON for the non-dust AOD. For the dust AOD, this coincides with the Harmattan season, which occurs from November through March. Reduced performance during the DJF season may indicate that the CMIP6 models are struggling to replicate the seasonal cycle of circulation and associated dust changes in the Harmattan season. Poor performance in the Harmattan season makes projections of changes in air qual-

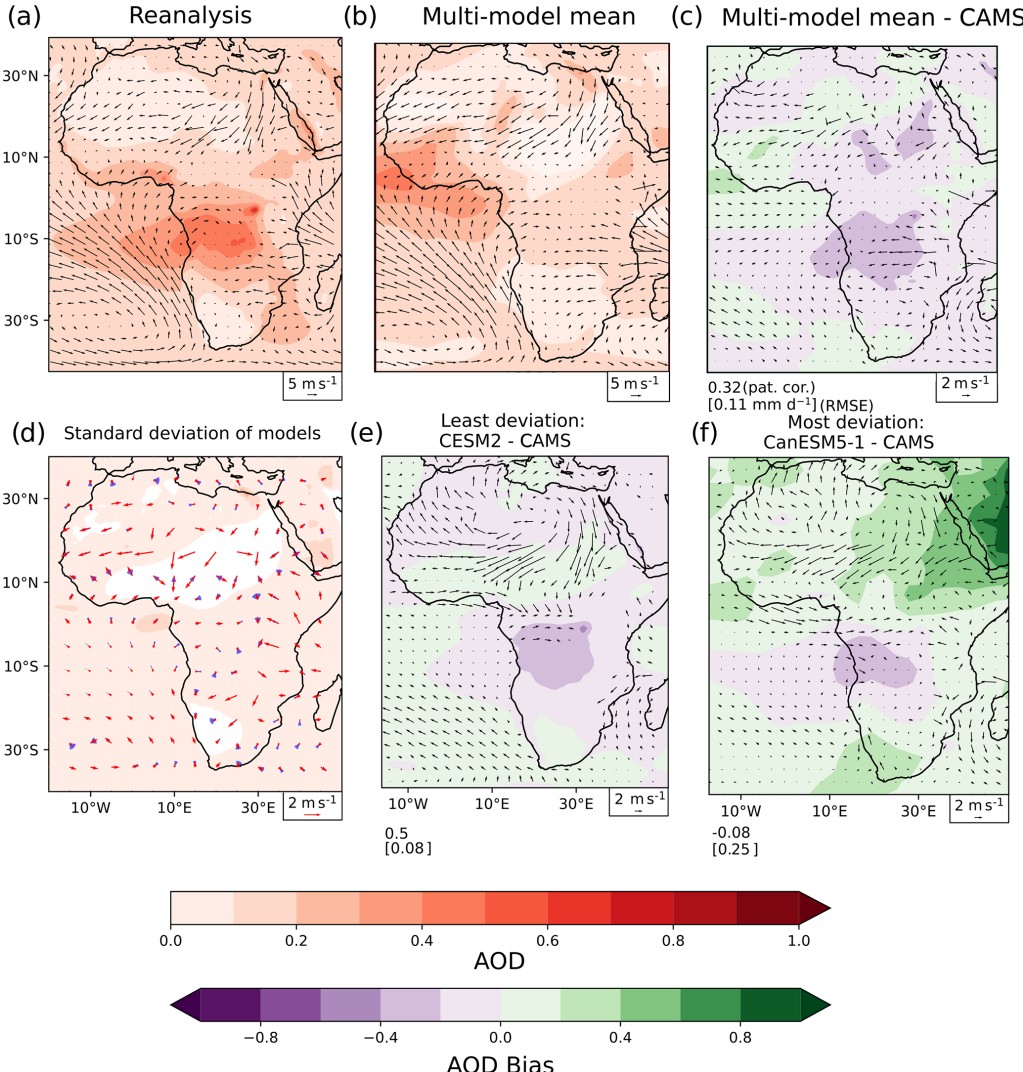

**Figure 5.** Same as Fig. 4 but for non-dust AOD. The model with the least deviation from CAMS is now CESM2, and the model with the most deviation from CAMS is now CanESM5-1.

ity, especially extreme air quality events, from the CMIP6 ensemble less reliable. For the non-dust AOD, SON is associated with biomass burning in central Africa, where the strong negative bias is seen in all of the models. Emissions inventories may not be capturing the magnitude of biomass burning emissions over the central African region during SON, as current literature notes underestimates of AOD increases in southern Africa (Lund et al., 2023).

Inaccuracies in emissions inventories may be the cause of the poorer performance for non-dust AOD over Africa. However, there is higher diversity in the pattern correlation coefficients for non-dust aerosol. Therefore, differences in the treatment of aerosols by different models still drive the strong intermodel diversity even for prescribed aerosol emissions.

### 3.3.1 West African AOD climatology

Different regions in Africa are characterised by different AOD climatologies according to relevant emissions and local meteorology, with consequences for human health and local climate. Understanding the impacts of evolving aerosol emissions is reliant on accurate interactions of the emissions with the meteorology of their source regions.

The normalised root mean square error (NRMSE) is used to rank model performance from least deviation (lowest NRMSE) to most deviation (highest NRMSE).

Figure 7 shows the seasonal cycle of AOD over west Africa in the reanalysis (CAMS), MMM, the model with the lowest NRMSE (MRI-ESM2-0), and the model with the highest NRMSE (MIROC-ES2L). The interannual variability of the seasonal cycle is shown by the shaded region.

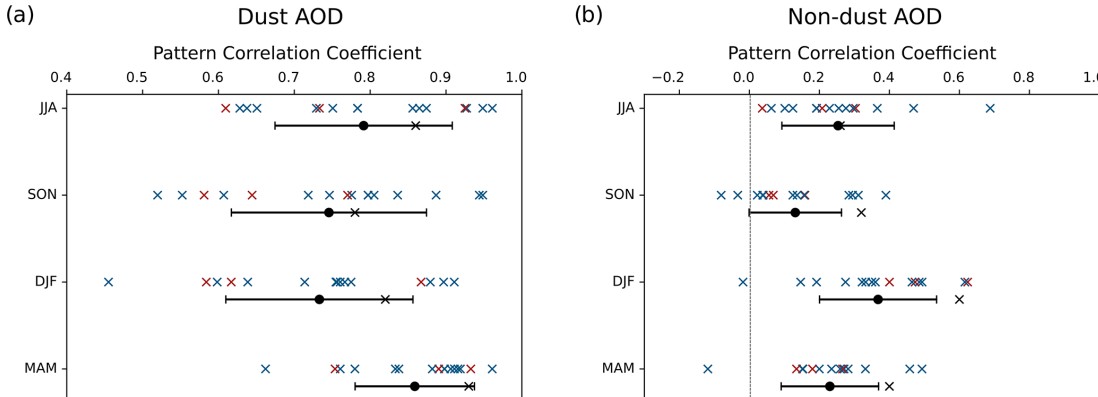

**Figure 6.** Models ranked by pattern correlation for seasonal **(a)** dust AOD and **(b)** non-dust AOD over Africa (40° S–40° N, 20–50° E), with the RAMIP models highlighted in red. Black dots show the mean of the individual model pattern correlation coefficients, black crosses show the MMM pattern correlation coefficient, and whiskers show the standard deviation of the model pattern correlation coefficients. The evaluation is performed using CAMS over the time period 2003–2023.

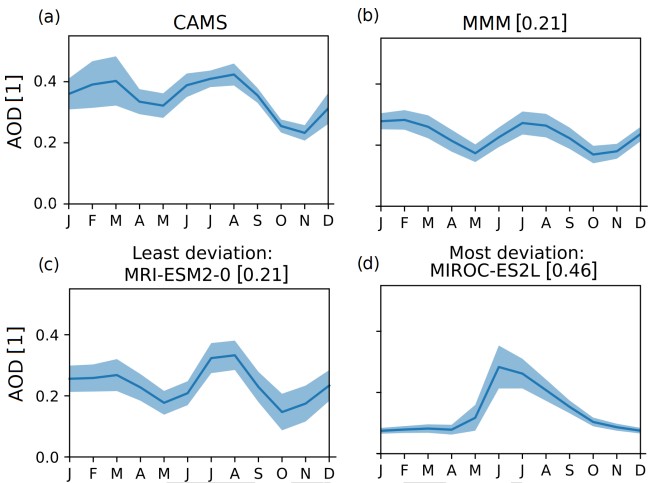

**Figure 7.** Monthly mean total AOD over west Africa (10° S–15° N, 20° W–25° E) for **(a)** reanalysis (CAMS), **(b)** MMM, **(c)** the model with the lowest NRMSE (MRI-ESM2-0), and **(d)** that with the highest NRMSE (MIROC-ES2L), with NRMSEs against CAMS shown in square brackets. Shaded region shows the interannual standard deviation for each month. The evaluation is performed using CAMS over the time period 2003–2023.

In Fig. 7a, AOD can be seen to have two peaks in west Africa in February–March and JJA. This climatology is driven by seasonal winds bringing dust from the Sahara, local aerosol emission annual cycles, and monsoon seasons of high rainfall reducing AOD. The peak from December to March (DJFM) is due to Harmattan winds bringing dry dusty air from the Sahara, while Senghor et al. (2017) links the peak in JJA AOD to transport from coastal sand sources. For the interannual variability, higher variability is noticeable during the Harmattan season, DJFM, which stems from differing strength of the Harmattan winds interannually – this can

be seen in the high standard deviation of lower-tropospheric wind speed during DJFM compared to the rest of the year.

The MMM shown in Fig. 7b shows the two peaks, though there are issues with the timing of the peaks; the first peak occurs too early by 1 month. The change in interannual variability in AOD through the year is not captured either, so the variability of the Harmattan season is not captured. Overall, the MMM captures the pattern of AOD climatology to the first order, though it does not correctly capture the timing of the AOD peaks and does not capture the changes in variability throughout the year.

The model with the lowest NRMSE, MRI-ESM2-0, better captures the distinct traits of the AOD peaks in each season. There are still issues with the timing of the peaks – while the first peak occurs at the right time, the second peak is too late and at least a month too short. In addition, the variability is similar year round, with a slight increase in October–November, which is the opposite of the variability seen in CAMS. These biases could originate from issues with capturing interannual variability of the Harmattan season, as the zonal mean precipitation climatology is found to be accurate, with only small wet biases during March, April, and May (MAM). Examining the interannual standard deviation in seasonal wind speed over Africa in MRI-ESM2-0 shows that, while capturing some increases in variability during DJF, the area of increased variability is too small, and the increase is too weak to fully capture the Harmattan and its effects.

The model with the highest NRMSE, MIROC-ES2L, does not capture the two-peaked climatology at all, instead having a single AOD peak in June–July. In addition, the variability in this model is strongest in the June–July period and is minimal elsewhere. This could relate to issues in the precipitation distribution, as MIROC-ES2L is found to have strong wet biases over west Africa – if a model has too many wet days, the lifetime of aerosol can be drastically reduced due to

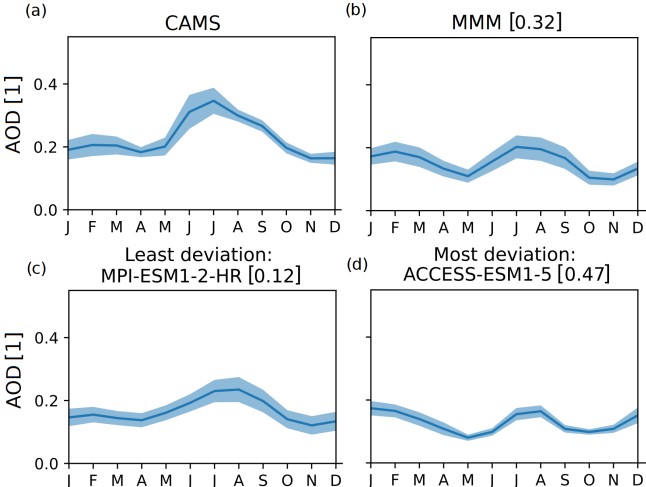

**Figure 8.** Monthly mean total AOD over east Africa (5° S–15° N, 27–46° E) for **(a)** reanalysis (CAMS), **(b)** MMM, **(c)** the model with the lowest NRMSE (MPI-ESM1-2-HR), and **(d)** that with the highest NRMSE (ACCESS-ESM1-5), with NRMSEs against CAMS shown in square brackets. Shaded region shows the interannual standard deviation for each month. The evaluation is performed using CAMS over the time period 2003–2023.

excessive scavenging, causing a low bias in AOD. The latitudinal progression of rainfall in MIROC-ES2L shows a strong wet bias in JAS, but the modelled AOD is most accurate during JAS, and so there are no obvious parallels between the poor performances in both the AOD and the zonal mean precipitation climatology.

Overall, the general pattern of AOD climatology throughout the year is captured, though there are some issues with the timing of peak AOD. Interannual variability in AOD associated with the Harmattan season is found to be too low in the MMM and the best-performing model, with both failing to capture the higher variability during this season. These issues with AOD climatology could be related to the biases in circulation (particularly the Harmattan winds that cause the DJFM AOD peak) and precipitation behaviour over the region, particularly biases in the number of wet days per year.

### 3.3.2 East African AOD climatology

Figure 8 shows the climatology of AOD over east Africa in the reanalysis dataset, CAMS, the AOD climatology of the MMM, and the AOD climatologies of the models with the lowest and highest NRMSE.

In Fig. 8a, the AOD climatology characteristics differ greatly to that of west Africa. Dust from the Sahara Desert is not as dominant as for west Africa, and interannual variability is reduced. The AOD can be seen to have one main peak in JJA. This is due to an increased positive zonal wind speed during JJA, with westerlies bringing in pollution from central Africa, compared to the rest of the year, in which in-

creased easterly winds bring cleaner air from over the oceans (Hastenrath et al., 2011). For the interannual variability, there is little change during the year.

From Fig. 8b, the MMM can be seen to match this pattern poorly – the JJA peak in AOD is not well produced, and instead the models simulate a two-peaked climatology, more similar to that of west Africa. The variability remains constant throughout the year, similar to the reanalysis.

The model with the lowest NRMSE, MPI-ESM1-2-HR, represents the AOD climatology more accurately than the MMM. It features the strong peak in AOD during JJA, though this peak lags a month behind that of CAMS – similar to the zonal mean precipitation cycle (Sect. 3.4.1) of the model – and has a lower peak magnitude.

The model with the highest NRMSE, ACCESS-ESM1-5, has two peaks and fails to capture the strength of the AOD peak in JJA, as well as the length of this peak, which may relate to wet biases found in the zonal mean precipitation climatology for ACCESS-ESM1-5, which are strongest in SON. In addition, the zonal mean precipitation lags behind the observations by a month for this model, which may explain negative biases after the long rainy season. The variability for ACCESS-ESM1-5 is much lower than that for CAMS.

The climatology of AOD over east Africa is not represented as well as over west Africa, with the MMM failing to capture the magnitude of the JJA peak in AOD found in the reanalysis. Interannual variability here is better represented than that over west Africa, though east Africa is not associated with the complex wind behaviour during the Harmattan season, so the climatology is not as complex. Biases over east Africa are found to relate to biases in the zonal mean precipitation cycle and may also relate to difficulties with circulation over the region, as this governs much of the seasonal cycle of AOD over east Africa.

### 3.4 Precipitation

For the observational uncertainty, African precipitation observational and reanalysis datasets demonstrate strong agreement over south and west Africa (Ayugi et al., 2024; Karypidou et al., 2022), though there are disagreements in precipitation climatology over east Africa and some differences in the behaviour of daily precipitation (Sylla et al., 2013). Difficulties in constraining precipitation observational uncertainties over east Africa are mainly due to a scarcity of rain gauge observations (Dinku et al., 2018). An example of the diversity in east African rainfall climatology is shown in Figure 9, through the latitude-time progression of the east African monsoon. It can be seen that the observational datasets disagree over both the northward extent and magnitude of mean rainfall over the region, though the annual cycle itself is consistent – the banded appearance of the ERA5 dataset is due to small hotspots of precipitation over east Africa caused by local orography. Precipitation over the majority of Africa

in CHIRPS has been found to be reliable (Dinku et al., 2018); however, the evaluation of model performance over east Africa should be understood in the context of the observational uncertainty.

Figure 10 shows the observational distribution of rainfall during JJA from CHIRPS, as well as the lower-tropospheric winds from ERA5, the CMIP6 multi-model mean (MMM), the intermodel standard deviation, the bias in the MMM relative to CHIRPS, and the biases in the models with the least and most deviation from the rainfall in CHIRPS, based on a pattern correlation approach. JJA is used as the example season for the precipitation analysis, as it shows the strongest intermodel diversity and lowest overall performance.

The most prominent feature seen in Fig. 10 is the ITCZ: the region of intense rainfall centred around 10° N in JJA. Capturing the seasonal and interannual position changes of the ITCZ is important over Africa, as it has a strong influence on both the WAM and EAM. Historically, sustained equatorward shifts in the northward extent of the ITCZ caused by remote aerosol emission changes have led to strong droughts over the west African region (Monerie et al., 2023). Therefore, capturing the correct position and inland extent of the ITCZ for each season is highly important.

Figure 10 shows that rainfall biases in JJA are largely confined to the region associated with the ITCZ – it can also be seen that these are the areas with the largest intermodel spread. Though the MMM captures the magnitude and eastward extent of the WAM well, there is a southward bias in the ITCZ over west Africa, a well-known bias that has persisted over generations of CMIP models (Bock et al., 2020); the continued presence of this bias in CMIP3, CMIP5, CMIP6, and, moreover, in HighResMIP experiments demonstrates that increasing model resolution in parameterised models does not remove this bias. However, this analysis did not identify any bias in the pattern of seasonal AOD relating to this positional bias in seasonal rainfall. The wind biases for the MMM in Fig. 10 show that the Saharan heat low (SHL) has a small ($< 5°$) southward bias. The SHL impacts the intensity and location of the WAM (Lavaysse et al., 2016). Therefore, this could be a cause of the southward bias in the WAM.

In the MMM during JJA, west Africa generally exhibits stronger biases than east Africa, with localised areas of wet biases such as over Nigeria and Gabon. In the MMM, a dry bias and hot spot of large intermodel spread can be seen over Ethiopia. This is over the Ethiopian highlands and may be due to orographical effects of the region not being captured because of low resolution limiting the area of high elevation.

The models analysed are found to perform well over the whole of Africa in JJA. Pattern correlations are found to be high for the MMM at 0.85, though there are some biases in the location and magnitude of rainfall over some regions.

The model with the least deviation from CHIRPS, NorESM2-LM, shows little bias in the position of the ITCZ during JJA and some small areas of localised biases over east

Africa. While still showing a southward bias in the position of the SHL, the spatial pattern of the WAM is well captured.

The model with the greatest deviation from CHIRPS, CNRM-ESM2-1, shows a strong bias in the position of the ITCZ over west Africa and a strong underestimation of rainfall over central and east Africa, which may be associated with poor simulation of the ITCZ. The SHL in this model also appears to be too weak, with easterly biases in wind direction on the eastern flank of the SHL.

Overall, Fig. 10 shows that there are large differences in the ability of the CMIP6 models to replicate African rainfall in JJA, especially over west Africa and Ethiopia. The strongest intermodel spread is found over west Africa, though biases in rainfall location are found over both east and west Africa. These biases can mainly be attributed to differences in the location and strength of the ITCZ and SHL during this season.

Figure 11 shows a ranking of the pattern correlations of individual models evaluated for each season. The season associated with the weakest overall model performance and highest intermodel spread is JJA, typically associated with high rainfall over west Africa. Despite having the weakest seasonal performance, the mean of the model pattern correlations for JJA is 0.83, and the multi-model mean pattern correlation is 0.85. A large number of models closer to the observations belong to the CESM family, indicating that these models perform well in simulating west African rainfall.

Figure 11 also shows the model performance during SON, DJF, and MAM. The other seasons exhibit lower intermodel spread in pattern correlation and stronger overall performance. For the equinoctial seasons, the model performance is slightly better than in JJA, with a mean of 0.87 for SON and 0.85 for MAM. For DJF, the performance overall is also better than that for JJA, with a pattern correlation of 0.88.

For all seasons, there is a steep drop-off in model skill for the more poorly performing models that is most pronounced in the solstitial seasons. The decline in model skill for select models during these seasons may indicate issues capturing the northward and southward extents of the ITCZ during JJA and DJF, respectively. Understanding where the biases in poorly performing models originate is integral to making decisions on using their results in future projections.

### 3.4.1 West African monsoon

Capturing the meridional progression of the monsoons is important for confidence in forecasts and climate projections for Africa. The progression is characterised by key monsoon characteristics, such as onset, duration, demise, and intensity, which are important to agricultural practices. The progression of the monsoon is dependent on the behaviours of several circulation systems, such as the ITCZ, tropical easterly jet, African easterly jet, and sub-tropical westerly jet (Niang et al., 2020). Poor performance in replicating the evolution of the monsoon throughout the year can point to difficulties

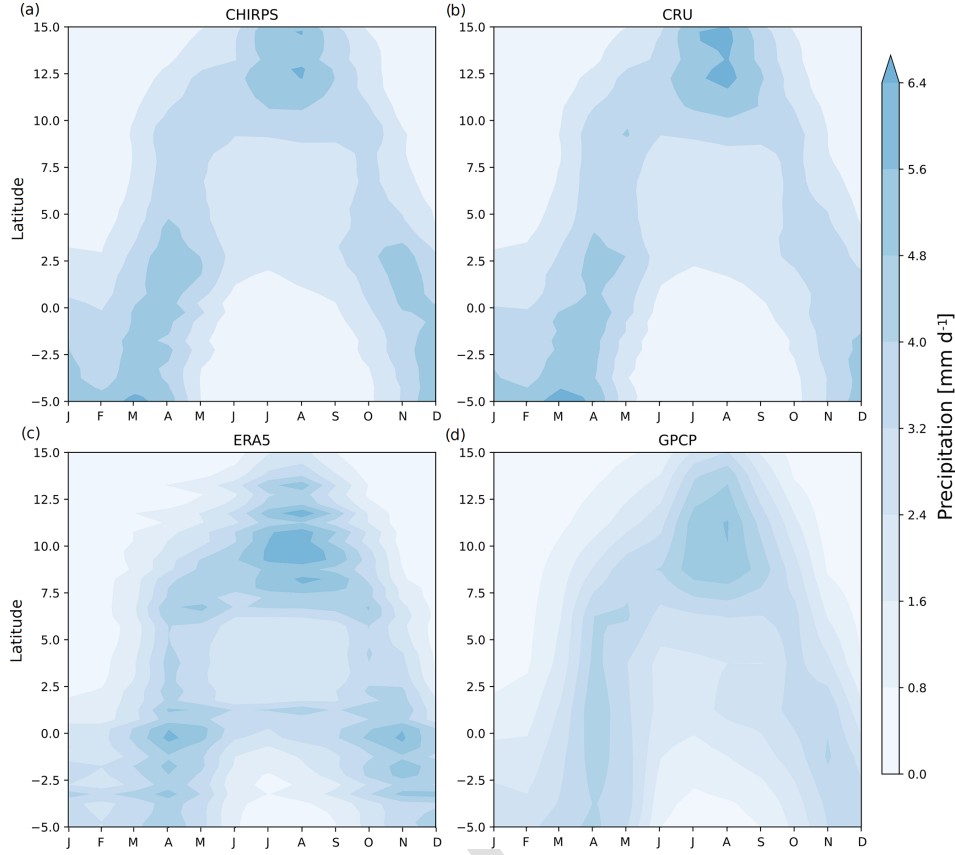

**Figure 9.** Observational estimates of precipitation in time-latitude diagrams of the tropical rain belt over east Africa for the **(a)** CHIRPS, **(b)** CRU, **(c)** ERA5, and **(d)** GPCP datasets over 1981–2023. A bounding box (5° S–15° N, 27–46° E) is used. Units are mm d$^{-1}$.

in the representation of these key circulation features. In addition, over west Africa in particular, the progression of the monsoon is dependent on local features, such as soil moisture gradients, so poor performance in the monsoon climatology can point to inaccuracy in local climate factors.

The latitudinal progression of the WAM throughout the year is shown in Fig. 12. The WAM is well captured, with a pattern correlation of 0.94 for the MMM and with individual models having patterns correlations in the range of 0.74–0.95. The temporal evolution of precipitation is well captured by the MMM, though there is an overall dry bias, as well a $\sim 4°$ southward bias in the peak rainfall (found through the difference in the latitudes of highest rainfall). The overall dry bias coincides with a negative AOD bias in the MMM for the AOD climatology over west Africa, though the opposite would be expected; with a drier monsoon season, less wet deposition of aerosols would lead to higher AOD.

The model with the highest pattern correlation, INM-CM4-8, underestimates the overall magnitude of rainfall, with a higher RMSE than the MMM, but captures the overall progression well, despite the southward bias in peak rainfall. The progression of intensity of the rainband for INM-CM4-8 lags behind that of CHIRPS, showing the strongest precip-

itation in October–November. No obvious relationships are found between the zonal mean precipitation shown here and AOD climatology, as the INM-CM4-8 AOD over west Africa is found to show a negative bias in AOD throughout the year.

The model with the strongest deviations from observations, UKESM1-0-LL, struggles to replicate the monsoon pattern over west Africa. This model shows the strongest rainfall for the region during MAM, much earlier than the peak in CHIRPS in August, which coincides with a negative bias in AOD over west Africa during MAM – potentially due to increased rates of wet deposition. In addition, the rainband becomes extremely weak in DJF compared to CHIRPS. This may contribute to the positive biases in AOD in UKESM1-0-LL over west Africa in DJF. The southward bias in the latitudinal location of peak rainfall is similar to that of the MMM. These biases may indicate difficulties with UKESM1-0-LL capturing the mechanisms governing the local monsoon evolution.

Overall, the WAM is well represented, especially in the MMM, despite a consistent southward bias, and overall agreement in the temporal rainfall pattern with CHIRPS indicates that the mechanisms governing the latitudinal progres-

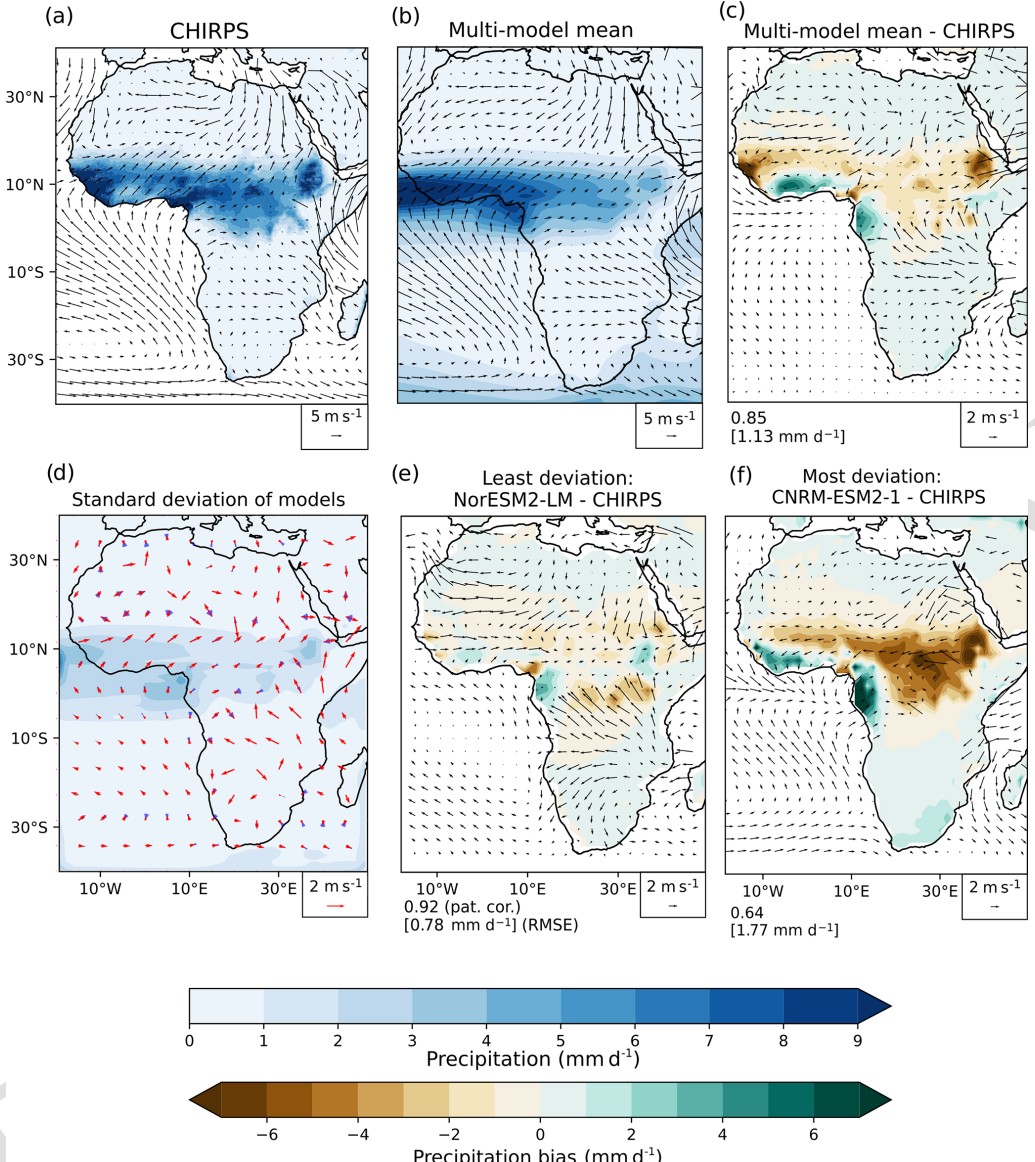

**Figure 10.** JJA mean rainfall and lower-tropospheric (925 hPa) winds for 1981–2023 in **(a)** observations (CHIRPS/ERA5) and **(b)** CMIP6 MMM. **(c)** CMIP6 MMM bias against observations and **(d)** intermodel standard deviation for precipitation (shading) and wind speed along the mean wind direction (red) and wind direction (blue). Mean rainfall and wind fields in the models with the **(e)** least (NorESM2-LM) and **(f)** most (CNRM-ESM2-1) deviation from CHIRPS as determined by pattern correlation. Pattern correlation and RMSE compared to CHIRPS are shown below panels **(c)**, **(e)**, and **(f)**. Note that the reference vectors for the 925 hPa winds differ between panels. Winds in panel **(d)** have been regridded to a $4° \times 4°$ grid for clarity.

sion of the monsoon are also well represented by the majority of models.

### 3.4.2   East African monsoon

The east African monsoon has received renewed attention due to ongoing severe droughts (World Health Organisation, 2024); simulating the east African monsoon is of no less importance than in west Africa, and thus monitoring and predicting changes in rainfall over this region in the near future

is vital for informing climate adaptation. Therefore, knowledge of the biases in rainfall over this region is also needed to use the CMIP6 models effectively in projections.

The latitudinal progression of the east African monsoon (EAM) throughout the year is shown in Fig. 13 for observations (CHIRPS), the MMM, the model with the least deviation from CHIRPS (MIROC6), and the model with the most deviation from CHIRPS (EC-Earth3).

Model performance in capturing the progression of the EAM is slightly weaker than that of the WAM. This is re-

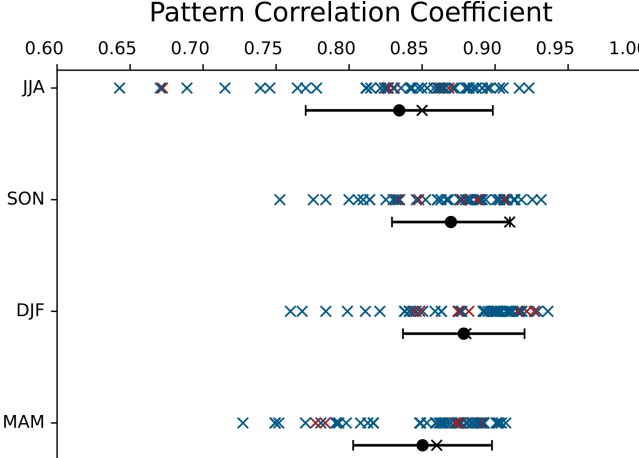

**Figure 11.** Models ranked by pattern correlation for seasonal rainfall over Africa (40° S–40° N, 20–50° E), with the RAMIP models highlighted in red. Black dots show the mean of the individual model pattern correlation coefficients, black crosses show the MMM pattern correlation coefficient, and whiskers show the standard deviation of the model pattern correlation coefficients. The evaluation is performed using CHIRPS over the time period 1981–2023.

flected in the CMIP6 MMM having a pattern correlation of 0.88 (compared to 0.94 for the WAM), while individual models have pattern correlations with CHIRPS ranging from 0.91 to as low as 0.36. The EAM MMM, unlike that of the WAM, shows no strong bias in overall precipitation magnitude and captures the northward extent of the rainband well. The evolution of rainfall is correct, though MAM (the long rainy season) is too dry compared to observations and October, November, and December (OND) (the short rainy season) has a wet bias. The biases found could relate to difficulties capturing the movement of ITCZ or impacts from the local orography being poorly represented (Munday et al., 2021, 2022).

The model with the least deviation from CHIRPS, MIROC6, is able to capture the time evolution of latitudinal progression in rainfall well. The model shows an overall wet bias common for MIROC6 over Africa – the RMSE associated with the model is higher than that of the MMM (1.34 mm d$^{-1}$ for MIROC6, 0.98 mm d$^{-1}$ for MMM). Overall, this model captures the latitudinal progression of rainfall over east Africa well, though the magnitude of mean rainfall is too high. This aligns with good performance in replicating the annual cycle in AOD over east Africa, reflecting good performance in latitudinal progression in rainfall. The AOD climatology also shows a negative bias throughout the year, which may relate to the positive bias in mean rainfall magnitude.

The model with the most deviation from CHIRPS, EC-Earth3, has very little resemblance to observations, with a pattern correlation of 0.36. The rainband, which in CHIRPS

has a maximum intensity in MAM and OND, can be seen to have a maximum only during OND, with no other season of strong rainfall. In addition, the spatial extent (5° S–∼ 10° N) is too large. The rainfall during this period is too strong, shown by an RMSE of 3.3 mm/day, to which the missing MAM wet season also contributes. The expected northward shift of the rainband in MAM is not present, and neither is the southward movement in SON.

Overall, the EAM is also well represented in the CMIP6 ensemble, though model performance is not as strong as for the WAM. While the absolute bias in mean precipitation is lower, the intermodel diversity is larger over this region than over west Africa, with large differences in pattern correlations and RMSEs for the models.

### 3.4.3   Daily precipitation

Beyond the overall temporal and spatial progression of rainfall patterns, it is important for climate models to be able to capture the characteristics of daily rainfall. As extreme events become more pronounced in a warmer world, being able to capture the extent of extreme precipitation and the correct distribution of rainfall amounts on wet days is integral to predicting the changes in extremes. Africa is highly vulnerable to flooding events. This is due to many regions experiencing positive trends in both severity and frequency of flooding (Tramblay and Villarini, 2020; Ekolu et al., 2022) and there being limited resources available to these areas for flood mitigation (Di Baldassarre et al., 2010). In addition, proper daily rainfall characterisation is important for the simulation of aerosol distributions and air quality (AOD and PM$_{2.5}$), as scavenging by rainfall acts to reduce aerosol lifetimes in the atmosphere. Rainfall that is too frequent, as is common in many global climate models (Emmenegger et al., 2024), is likely to lead to reduced aerosol burdens.

Figure 14 shows the number of wet days (days with > 2 mm d$^{-1}$ rainfall) per year over Africa for observations (CHIRPS), the MMM, and the MMM bias – individual model behaviour is examined in their daily precipitation performance. This again shows a common CMIP bias, where there are too many wet days compared to observations. This bias is strongest on the west central African coast and is substantial over this region, reaching over 160 extra wet days per year. This coincides with an area of low bias in annual mean AOD, though preliminary analysis does not show a strong relationship between increased number of wet days and decreasing AOD. The bias is also evident over areas of southern Africa and east Africa. The southward bias in rainfall is also evident through this plot. There is also a noticeable region over Uganda that has too few wet days, though the reasons for this bias over the region are not currently clear.

To look more closely at daily rainfall behaviour for the monsoon regions, Fig. 15 shows the probability density functions (PDFs) for daily rainfall over west and east Africa for CHIRPS, the MMM, and the models with the lowest and

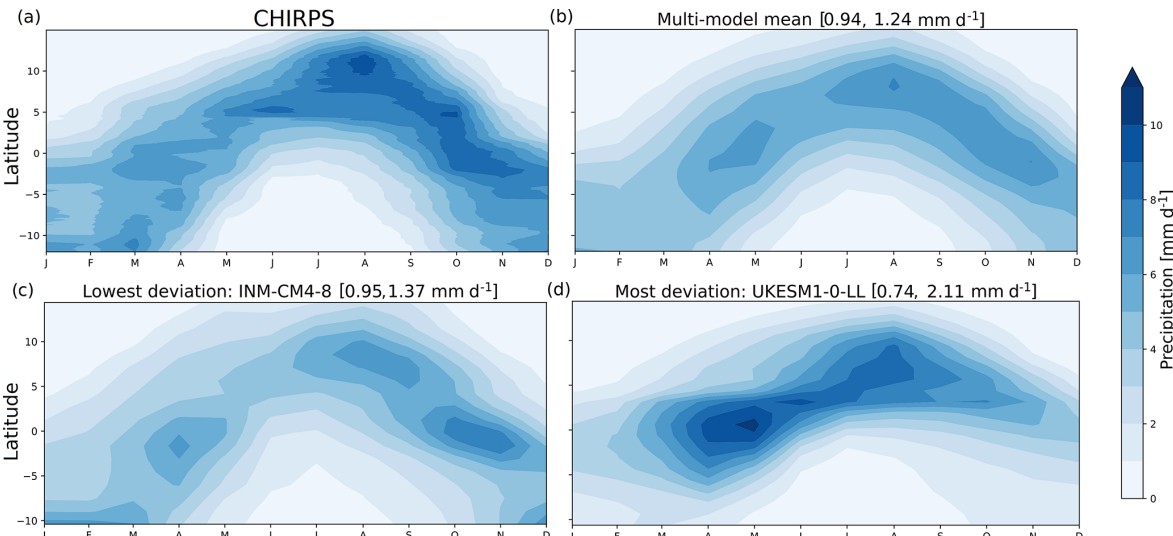

**Figure 12.** Latitudinal progression of the tropical rain belt for west Africa through the seasonal cycle, averaged over 1981 to 2023 for the **(a)** observations (CHIRPS), **(b)** MMM, **(c)** the model with the smallest deviation from observations (INM-CM4-8), and **(d)** the model with the greatest deviation from observations (UKESM1-0-LL), with pattern correlations and RMSEs shown in square brackets above each panel. The data shown cover 10° S–15° N, 20° W–25° E. Units are mm d$^{-1}$.

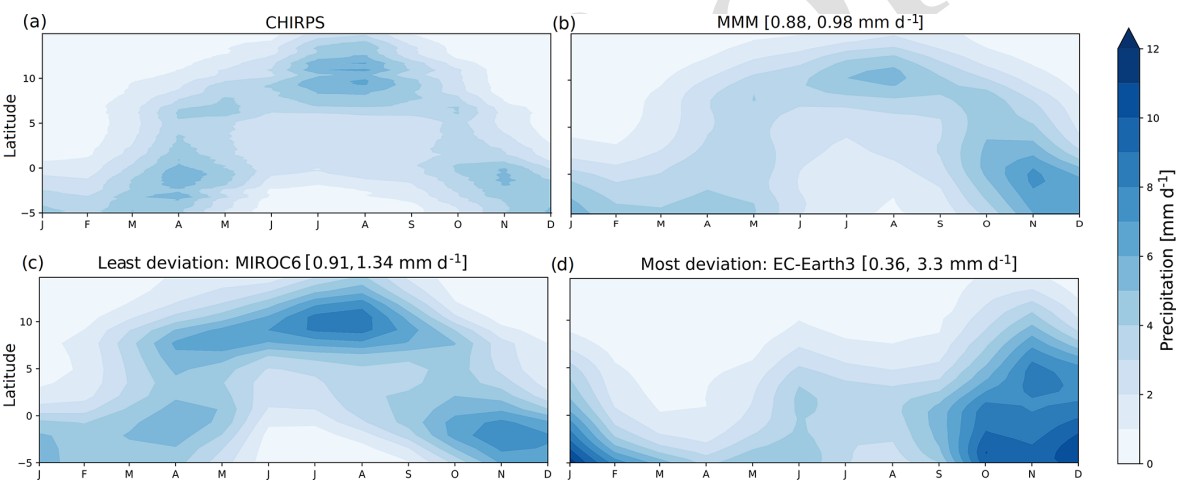

**Figure 13.** Latitudinal progression of the tropical rain belt for east Africa through the seasonal cycle, averaged over 1981 to 2023 for the **(a)** observations (CHIRPS), **(b)** MMM, **(c)** the model with the smallest deviation from observations (MIROC6), and **(d)** the model with the largest deviation from observations (EC-Earth3), with pattern correlations and RMSEs shown in square brackets above each panel. The data shown cover 5° S–15° N, 27–46° E. Units are mm d$^{-1}$.

highest RMSEs. The PDFs are calculated from regridded daily precipitation datasets in order to reduce the impact of differing resolutions of the different climate models and observations. Days with total rainfall of less than 2 mm are excluded from this section of the analysis to examine only the behaviour of rainfall on wet days.

As shown in Fig. 15a for west Africa, the observations (CHIRPS) show a large spread of daily precipitation values, with high extremes of over 50 mm d$^{-1}$ found in the region. In comparison, the PDFs of the MMM and highest RMSE model show that they produce too much drizzle, and the high-

est RMSE model fails to capture the days of very intense rainfall. This is in agreement with current literature that identifies biases with CMIP6 models producing too many days of light rainfall due to rainfall parameterisation over grid boxes (Emmenegger et al., 2024).

The model with the lowest RMSE, CAMS-CSM1-0, shows a wider range of daily precipitation values than the MMM and highest RMSE model. CAMS-CSM1-0 captures the correct frequency of days with light rain and only slightly underestimates the frequency of the high-precipitation days.

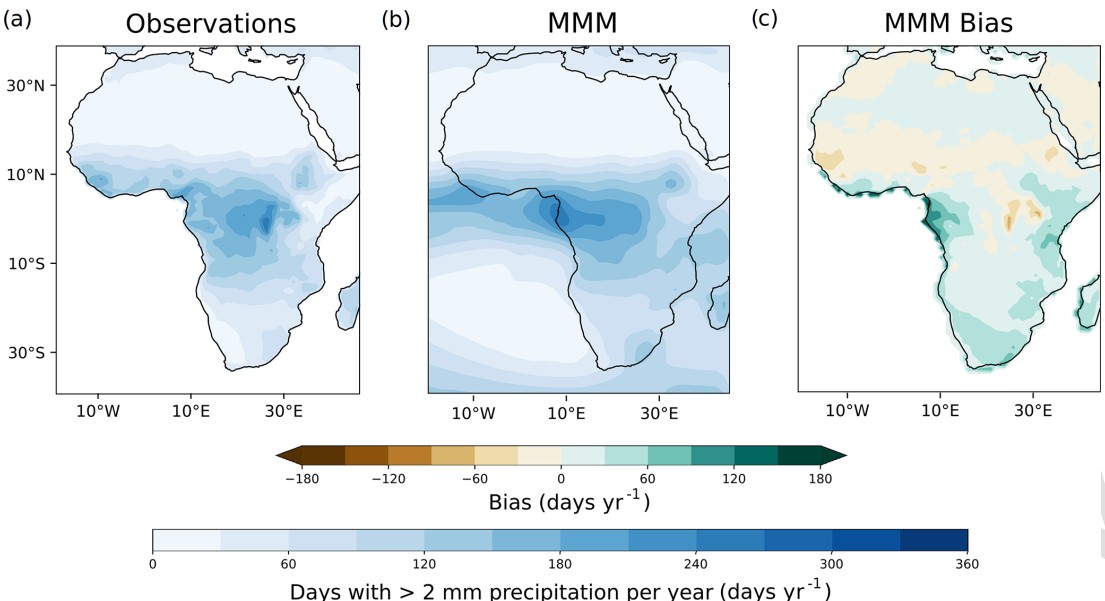

**Figure 14.** Number of wet days ($> 2\,\mathrm{mm\,d^{-1}}$) per year for **(a)** CHIRPS, **(b)** MMM, and **(c)** MMM bias over Africa. The data shown are from 1981 to 2023.

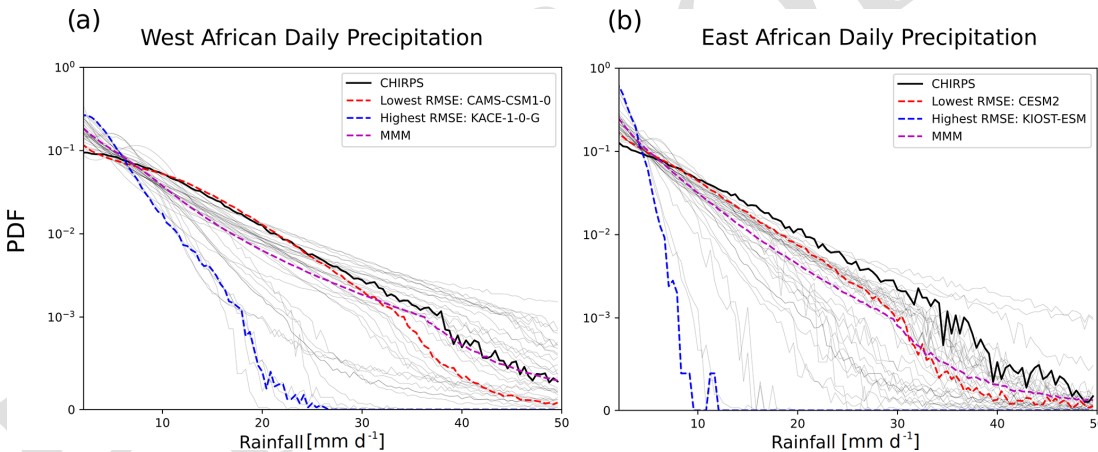

**Figure 15.** Probability density functions for daily rainfall on wet days (defined as $> 2\,\mathrm{mm\,d^{-1}}$) for each grid point over regions in west (10° S–15° N, 20° W–25° E) **(a)** and east (5° S–15° N, 27° E–46° E) **(b)** Africa. The multi-model mean of the distributions is shown, as well as the lowest RMSE model (CAMS-CSM1-0 for west Africa, CESM2 for east Africa), highest RMSE model (KACE-1-0-G for west Africa, KIOST-ESM for east Africa), and observations (CHIRPS). PDFs of all other models are shown by the light-grey lines. All models and observations have been regridded to a common 1° × 1° grid. The data shown are from 1981 to 2023.

The model with the highest RMSE, KACE-1-0-G, shows a low spread in daily precipitation values, with high frequencies associated with low daily precipitation values. In addition, the range of daily precipitation values for KACE-1-0-G extends only to 24 mm d$^{-1}$ – less than half of the range found in CHIRPS.

Overall, the daily precipitation over west Africa is well represented in most models, though there are issues stemming from drizzle associated with some models, and the extreme high daily precipitation values lack representation for some models.

As shown in Fig. 15b for east Africa, the observations show a slightly smaller spread in daily precipitation values, with maximum daily precipitation values of up to 50 mm d$^{-1}$. The shape of the PDF is similar to that of west Africa. Models over this region perform worse in replicating the behaviour of daily precipitation, especially in capturing the range of daily precipitation values.

The model with the lowest RMSE, CESM2, has a PDF showing a bias towards more days with less precipitation. Both CESM2 and the MMM fail to capture the frequency of days with intense rainfall.

The model with the highest RMSE, KIOST-ESM, has a strong bias towards days with low precipitation. The range of daily precipitation is extremely low in this model, up to 10 mm d$^{-1}$, showing that days of intense rainfall are not well captured by this model.

Overall, the PDF for daily precipitation is better represented over west Africa than east Africa, consistent with the representation of seasonal mean precipitation. Most models replicate the range of daily precipitation found in observations in both regions; however, a number of models produce too many days with light, but non-zero, precipitation.

## 4 Discussion and conclusions

We have evaluated the performance of CMIP6 models in simulating PM$_{2.5}$, aerosol optical depth (AOD), and precipitation over Africa, relative to observational and reanalysis products. PM$_{2.5}$ performance, which was evaluated against a novel observational dataset, was indicated using the $R^2$, root mean squared error (RMSE), and mean absolute error (MAE) metrics. For AOD and precipitation, we evaluated their performance through the use of pattern correlation coefficients and RMSEs for the seasonal mean rainfall and AOD, pattern correlation coefficients for the latitude-time progression of monsoon rainfall, and the normalised root mean squared error (NRMSE) for the seasonal cycle of AOD.

PM$_{2.5}$ concentrations derived from model mass mixing ratios are found to exhibit similar annual cycles to that of the AirNow dataset, with notable exceptions such as Cairo. However, time series over some of the stations examined are less than 1 year in length. Therefore, further investigation as more observational data become available is necessary to reliably characterise PM$_{2.5}$ behaviour for each observation station. Recent political developments leading to the termination of the AirNow dataset may prevent further exploration of relationships between air quality and precipitation in locations such as Cairo and Kigali. The majority of the models examined underestimate the PM$_{2.5}$ concentration, which could be due to comparing observations taken in urban areas to grid cell variables, models underestimating the number of dry days identified in this analysis, and missing processes in models, such as the modelling of nitrate.

Seasonal spatial patterns of dust AOD are fairly well represented, with DJF showing the strongest disagreement with reanalysis, due to eastward biases in the peak dust AOD from the Sahara Desert, with intermodel spread caused by differences in atmospheric circulation patterns, dust-emitting regions, and uncertainties in simulated dust emission rates due to differing parameterisations. Hotspots of intermodel disagreements over northeast Africa can be linked most strongly to differences in dust emission between models. Conversely, seasonal spatial patterns of non-dust AOD are more poorly represented, with SON being the season with the strongest disagreement with respect to reanalysis, especially in central Africa, where no models produce the area of high non-dust AOD found in CAMS, which may be due to underestimations in emissions inventories over the region. The annual cycle of AOD in CMIP6 is better represented over west Africa than east Africa, though the cycle of interannual variability is not well captured for west Africa. Underestimates in emissions inventories are a potential cause of non-dust AOD biases in the MMM during SON, especially over the biomass burning regions (Reddington et al., 2016), but do not fully account for the model spread found. Challenges in accurately modelling regional atmospheric circulation contribute to these discrepancies, especially model inability to correctly simulate the transport of biomass burning aerosols over central Africa. Further investigation of the regional biases would be possible through evaluation over smaller regions and examination of atmospheric circulation pattern performances across Africa.

There is intermodel diversity in the pattern of rainfall over Africa during JJA, but the seasonal rainfall patterns are well represented for the remaining seasons. For the CMIP6 models, though both are well represented, we find better performance overall in replicating the seasonal cycle of the west African monsoon than the east African monsoon. The origins of the rainfall biases over Africa may relate to biases in atmospheric circulation, such as the insufficient latitudinal progression of the ITCZ and biases in the wind bringing moisture from sea to land. Biases relating to the intertropical convergence zone (ITCZ) appear to have a stronger effect over east Africa, where the MMM shows some biases in the timing of the rainy seasons. In contrast, difficulties with overall rainfall magnitude are more influential over west Africa, where the MMM shows a dry bias in areas impacted by the tropical rainband. Schwarzwald et al. (2022) showed that while biases in SST have some impact on precipitation performance over east Africa, changes in precipitation performance made by prescribing SSTs only improve models with difficulties in recreating features of the Indian Ocean basin, so the biases found for precipitation are unlikely to originate solely from SST biases.

The differing performances of the models based on the season and region chosen underline the need to use a diverse range of models when aiming for robust insights into the future evolution of precipitation and AOD over Africa. This study has identified model performance over Africa as a key knowledge gap, and determining the climate response to local and remote aerosol emission is crucial for Africa; these knowledge gaps are closely linked. This evaluation will inform the analysis of modelled responses in RAMIP experiments, which investigate the impacts of local and remote aerosol emission changes on Africa. The results here inform the degree of confidence that can be placed in each model's responses based on their performance for the season and region of interest.

A number of instances where biases in AOD and PM$_{2.5}$ may relate to biases in precipitation, or vice versa, were identified in this analysis. Notable examples include poorer per-

formance in replicating the annual climatology of AOD over east Africa than over west Africa; this may relate to weaker model performance in replicating the latitudinal progression of rainfall throughout the year over east Africa compared to west Africa. In addition, MIROC6 and MIROC-ES2L were found to underestimate AOD and PM$_{2.5}$ in seasonal AOD distributions and PM$_{2.5}$ climatology, which may be linked to the strong positive biases in mean rainfall magnitude causing erroneously high rates of wet deposition of aerosol.

However, there were many areas where links could not be identified or where biases were opposite to those expected. For example, in NorESM2-LM a southward bias in dust AOD over west Africa coincided with good performance in seasonal rainfall pattern over the region, though problems with CESM2 family models were strongly linked to issues with emission location and rate, rather than transport or removal issues. Connections between dust AOD and surface wind speed over Africa were also not identified, despite being noted for some models in regions outside of and including Africa (Wu et al., 2021; Zhao et al., 2022). Therefore, the initial analysis suggests that wide diversity in dust AOD may be dominated by differences in dust schemes rather than differences in meteorological conditions for models.

A notable boundary in this analysis for comparing biases in AOD and rainfall is that dust and speciated AOD data are not available for many models, limiting our ability to identify impacts of inaccurate precipitation patterns on AOD behaviour.

Aerosol forcing has been identified as the largest contributor to uncertainty in anthropogenic effective radiative forcing (Myhre et al., 2013b), and it has been shown in previous studies that African precipitation is affected by aerosol emissions, from both local and remote sources (Monerie et al., 2023; Scannell et al., 2019; Shindell et al., 2023). CMIP6 models exhibit a large intermodel spread and biases in their simulations of air quality characteristics and precipitation patterns. Therefore, when coupled with high uncertainty in future projections of African aerosol emissions, this intermodel spread results in a poorly constrained outlook for the future evolution of sub-Saharan air quality and precipitation. These uncertainties and disagreements with observations, alongside vulnerability to climate and air quality changes, make it challenging to accurately quantify the impacts of these changes as global warming increases and emissions of short-lived climate forcers change. It is necessary to advise policy makers on emissions mitigation and local adaptation measures with a clear understanding of projections and model responses. Therefore, we highlight model performance over sub-Saharan Africa and the underlying reasons for these biases as crucial knowledge gaps in atmospheric science.

**Code and data availability.** This work uses simulations from 56 models participating in the CMIP project as part of the Coupled Model Intercomparison Project (Phase 6; https://esgf-ui.ceda. ac.uk/cog/search/cmip6-ceda/, WCRP, 2025); model-specific information can be found through the references listed in Table 2. Model outputs are available on the Earth System Grid Federation (ESGF) website (https://esgf-ui.ceda.ac.uk/cog/search/ cmip6-ceda/, WCRP, 2025; Cinquini et al., 2014). The reanalysis and observational data used in this work are all cited and publicly available. The analysis was carried out using the Bash and Python programming languages. The ERA5 reanalysis dataset was retrieved from ECMWF's Meteorological Archival and Retrieval System (MARS). See https://www.ecmwf.int/en/forecasts/ dataset/ecmwf-reanalysis-v5 (ECMWF, 2025) for further details. The CHIRPS dataset can be retrieved from the Climate Hazard Centre's Early Warning Explorer. See https://earlywarning.usgs.gov/ fews/ewx/index.html (CHC, 2025) (CHC, 2025) for further details. The AirNow dataset used for this study is no longer online, but a repository is available upon request, and a more limited version of the dataset is available at AirNow DOS Embassy and Consulate Monitoring Site Data (Last 24 h – PM$_{2.5}$ Only), US EPA, OAR, OAQPS – Overview https://www.arcgis.com/home/item.html?id= db1a45b351164464ac0459a52890e5a0 (AirNow, 2025).

**Supplement.** The supplement related to this article is available online at [the link will be implemented upon publication].

**Author contributions.** LJW, BHS, AGT, CAT, DMW, and JAA designed the study. JAA performed the data analysis and produced the figure for the PM$_{2.5}$ results and provided the discussion on PM$_{2.5}$. CAT performed the data analysis and produced the figures for the AOD and precipitation results and discussed these results. All authors edited the paper.

**Competing interests.** At least one of the (co-)authors is a member of the editorial board of *Atmospheric Chemistry and Physics*. The peer-review process was guided by an independent editor, and the authors also have no other competing interests to declare.

**Disclaimer.** Publisher's note: Copernicus Publications remains neutral with regard to jurisdictional claims made in the text, published maps, institutional affiliations, or any other geographical representation in this paper. While Copernicus Publications makes every effort to include appropriate place names, the final responsibility lies with the authors.

**Acknowledgements.** We acknowledge the World Climate Research Programme, which, through its Working Group on Coupled Modelling, coordinated and promoted CMIP6. We thank the climate modelling groups for producing and making available their model output, the Earth System Grid Federation (ESGF) for archiving the data and providing access, and the multiple funding agencies who support CMIP6 and ESGF. We acknowledge the use of ERA5 data produced by ECMWF, CHIRPS data produced by the Climate Hazards Center, and CAMS data produced by ECMWF. Additional details regarding ERA5 and CAMS can be found at https://cds. climate.copernicus.eu/ (last access: 26 September 2024), CHIRPS

at https://www.chc.ucsb.edu/data/chirps (last access: 30 September 2024), and CAMS at https://ads.atmosphere.copernicus.eu (last access: 21 August 2025). The analysis in this work was performed on the JASMIN super-data cluster (Lawrence et al., 2012). JASMIN is managed and delivered by the UK Science and Technology Facilities Council (STFC) Centre for Environmental Data Analysis (CEDA).

**Financial support.** Laura J. Wilcox and Andrew G. Turner are supported by the Natural Environment Research Council (NERC; grant NE/W004895/1, TerraFIRMA) and the National Centre for Atmospheric Science. We acknowledge the Centre for Advanced Study in Oslo, Norway, which funded and hosted our HETCLIF centre during the academic year of 2023/2024. Daniel M. Westervelt and Joe Adabouk Amooli are supported by the U.S. National Science Foundation Office of International Science and Engineering Grant 2020677. Bjørn H. Samset acknowledges funding from the Research Council of Norway, Grant 324182 (CATHY). Catherine A. Toolan acknowledges PhD studentship funding from the SCENARIO Natural Environment Research Council (NERC) Doctoral Training Partnership grant and a CICERO CASE studentship.

**Review statement.** This paper was edited by Philip Stier and reviewed by three anonymous referees.

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
