# Peer review of "Strong inter-model differences and biases in CMIP6 simulations of $PM_{2.5}$ , aerosol optical depth, and precipitation over Africa"

_EGUsphere, 2024_

## Author Response (AR1)

We thank the reviewers for their helpful comments for improving our manuscript. The reviewer comments are shown in **bold**, with responses below in standard font, and lines noted in the latexdiff manuscript where relevant changes have been made in *italics.*

**REVIEWER 1:**

**Toolan et al. evaluate the performance of CMIP6 models in simulating PM2.5, aerosol optical depth (AOD), and precipitation over Africa. The authors provide a thorough analysis using observational and reanalysis datasets, identifying regions and variables where models perform well and where they require improvement. This study has potential to contribute significantly to understanding the reliability and limitations of CMIP6 models for regional applications in Africa. I recommend the manuscript for publication in Atmospheric Chemistry and Physics, subject to the following major and minor comments.**

We thank the reviewer for their positive summary of our study and their recommendation for publication, and are pleased with their recognition of the study's contribution to understanding CMIP6 models.

**Major Comments**

**The analysis highlights model biases and performance differences across multiple variables and regions in CMIP6 models, but its scientific contribution and novelty could be improved by addressing certain gaps in linking these variables. Currently, the explanations regarding PM2.5, AOD, and precipitation are treated somewhat independently, and the connections between them are not explicitly discussed. For instance, how systematic discrepancies in the ITCZ influence aerosol transport / air quality and precipitation variability. This exploration could provide essential insights into understanding and identifying (or rule out) the key physical processes to drive the inter-model uncertainties.**

Discussion linking the three variables evaluated has been expanded in Section 1 - Introduction, where we outline mechanisms for interaction *(lines 58-68)*. Section 3 - Results now also includes further discussion of biases in the latitudinal position of rainfall and precipitation magnitude may have influenced the biases found in air quality, through changes to aerosol transport and precipitation variability, mostly over east Africa *(line 570)*. There is also similar added discussion linking biases in precipitation with those found in air quality *(lines 310-322)*. Some additional discussion of mechanisms of dust AOD biases have been added to Section 3.4 - Precipitation *(line 535)*. Further discussion linking precipitation distribution to PM2.5 concentration is also added to Section 3.2- PM2.5 *(lines 310-322)*.

We also discuss the differences in relationships between precipitation and PM2.5 levels between the east and west African AirNow sites evaluated, demonstrating the complexity of interactions between the two *(lines 315-322)*. To aid this, we have added a figure to the supplementary material showing timeseries of CHIRPS daily rainfall measurements against PM2.5 levels for each site. There are limitations in the comparisons of the datasets, since the AirNow dataset is for individual cities, and CHIRPS does not resolve at a city-scale. This limits direct comparison but broad regional differences in air quality/precipitation relationships between east and west Africa can be seen.

**The discussion on AOD performance across models is detailed and robust, especially in separating dust and non-dust components. However, it could benefit more from some discussions on following climate impacts, such as aerosol-radiative and aerosol-cloud interactions in the CMIP6 climate models (which have been previously shown to largely affect African climate but with large uncertainties), to further strengthen the impacts of this work.**

We have expanded the introduction to note impacts of aerosol-radiative and aerosol-cloud interactions in CMIP6 models. This includes discussion on:

- Differences in aerosol optical properties and microphysics schemes across models contributing to uncertainties in aerosol forcing (direct and indirect) and so climate responses over Africa *(lines 46 and 59)*. References from RC2 were also helpful in addressing this point by demonstrating uncertainty in aerosol effective radiative forcing.
- Wider implications of these interactions, noting, among others, Shindell et al (2012), Shonk et al (2020) and Williams et al (2022), which linked aerosol forcing to regional precipitation patterns and monsoon dynamics *(lines 697-699)*.

Our mention of climate impacts of aerosol-radiative and aerosol-cloud interactions was also reworked to demonstrate their relevance to CMIP6 model evaluation and African climate projections *(lines 59 and 67)*.

**I have some questions on the motivation of the choice of CMIP6 models for evaluating air pollution, due to their coarse resolution and relatively simplified chemical schemes. While these models are suitable for large-scale climate studies, their limitations in capturing small-scale and chemical processes critical for air quality should be explicitly acknowledged. A discussion on how model resolution and the simplified chemical schemes affects the results, and whether higher-resolution regional models like WRF-Chem might be more appropriate for certain aspects of this analysis, would further strengthen the manuscript.**

Clarification has been added to acknowledge intrinsic limitations for global climate models in capturing air quality behaviour (in Section 1 - Introduction), as well as further discussion on PM2.5 results in the context of differing resolutions and chemistry schemes. However,

CMIP6 models provide an opportunity to link air quality to large-scale circulation changes, which isn't possible with regional models. They are used for air quality studies (e.g. Turnock et al., 2020, Guo et al., 2021) and our evaluation is intended to serve this community. We now include a clear statement of this aim and motivation, alongside a discussion of the relative benefits of different model types, in the Introduction *(lines 127-132)*.

**Minor Comments:**

**Line 12: "well captured" is vague. Specify whether this refers to precipitation magnitude, variability, monsoon onset time, or other variables.**

Changed to 'the magnitude and annual progression of precipitation over both monsoon regions is well captured'.

**Line 18: "Africa is a region of large heterogeneity in both air quality and precipitation", 'heterogeneity' refers to spatial heterogeneity?**

'Spatial' added for clarity.

**Line 139: Specify the time period and resolution of CAMS data.**

Added.

**Section 2.1.3: Specify the time period, and resolution. Also, I think the section should be 'precipitation observation and reanalysis datasets' as ERA5 is not an observational data set.**

Added and changed.

**Line 202: provide the definition of the acronym "ITCZ" when it is first mentioned.**

Added.

**Line 210: Define MMR (mass mixing ratio) as it is first time mentioned here.**

Added.

**Line 223: How is pattern correlation calculated? For example, convert the time-averaged 2d date into 1d and then calculate the Pearson correlation coefficient?**

Yes, the data is time-averaged to 2d, then the rows are concatenated to make a 1d dataset, and the Pearson correlation coefficient is calculated. Added to the text for clarity.

**Figure 2 caption: NRMSE is not defined until section 3.3.1**

Definition added.

**Line 293-294: The explanation is a bit simplified here. Different model schemes can produce varying dust emissions, but the situation is more complex because the dust emission is interactively calculated. For instance, even when two models employ the same dust scheme, variations in simulated meteorological conditions (e.g., wind speed, soil moisture, and atmospheric stability) can result in very different dust emissions. It would be helpful to clarify (or comment on) whether the discrepancies arise primarily from differences in the dust schemes themselves, or whether they are predominantly driven by meteorological variability across the models.**

The central Saharan region has shown intermodel diversity in wind speed, so this is also likely to be a cause of the differences in dust AOD. Reflecting this, we've changed the line to: "Strong intermodel variability over the central Saharan region may be due to different interactive dust schemes used by many CMIP6 generation models, causing strong differences in the magnitude of dust aerosol. There is also some intermodel disagreement in wind speed over the region; as interactive dust emissions rely on surface wind speeds, this will contribute to differences in interactive dust emissions. Intermodel differences in other meteorological conditions which are not shown here, such as soil moisture, will also contribute to the diversity seen (Zhao et al, 2023)." Further discussion of dust AOD bias origins in CMIP6 models has been added to Section 1 - Introduction *(lines 74-86).*

**Line 306: overestimated in all models?**

Not all models, and the bias in the MMM is mainly driven by the CESM-family models discussed in the paragraph above. Rephrased for clarity.

**Line 567-570: This sentence is long and hard to read. Suggest to break down or rephrase.**

Rephrased to "PM2.5 concentrations derived from model mass mixing ratios are found to exhibit similar annual cycles to that of the AirNow dataset, with notable exceptions such as Cairo. However, timeseries over some of the stations examined are less than 1 year in length. Therefore, further investigation as more observational data become available is necessary to reliably characterise PM2.5 behaviour for each observation station."

**Line 582:The phrase "Difficulties in replicating the circulation over these regions are also responsible for some issues" could be more precise, e.g., "Challenges in accurately modeling regional atmospheric circulation contribute to these discrepancies"**

Suggested wording used.

**Generally I would suggest avoiding vague language like "not well captured", which repeatedly appears in the manuscript several times. Instead, specify the what is the**

**discrepancy, for instance "models fail to replicate the observed seasonal peak in non-dust AOD over central Africa during SON"**

Rectified where necessary throughout, e.g.:

Line 373: "Interannual variability in AOD associated with the Harmattan season is not well captured in the MMM or the best performing model." is changed to: "Interannual variability in AOD associated with the Harmattan season is found to be too low in the MMM and the best performing model, with both failing to capture the peak in variability during the season."

**REVIEWER 2:**

**Toolan et al. investigate how CMIP6 models perform in simulating PM2.5 levels, aerosol optical depth (AOD), and precipitation over Africa, focusing on east and west Africa. Overall, 56 CMIP6 models are evaluated relative to observational and reanalysis datasets on annual and seasonal scale during the 1981-2023 period. PM2.5 is evaluated relative to the AirNow database, dust and non-dust AOD is evaluated separately relative to CAMS, while precipitation is examined on monthly and daily basis to assess whether CMIP6 models accurately capture precipitation patterns throughout the year over east and west Africa, with focus on the east and west African monsoons. The authors use a variety of metrics to evaluate the performance of CMIP6 models and focus on interpreting their results throughout the manuscript. The results are comprehensive and advance our understanding of the weaknesses of climate models in accurately reproducing the climate state over Africa.**

We thank the reviewer for their appreciation of our work - we're glad they found the study comprehensive and believe that it contributes to understanding the limitations of climate models over Africa.

**Major Comments**

1. **Lines 44-45: You may also consider a number of recent studies that quantify the effective radiative forcing of anthropogenic aerosols and its decomposition in aerosol-cloud interactions and aerosol-radiation interactions using CMIP6 Earth system models (e.g., Thornhill et al., 2021; Zelinka et al., 2023; Kalisoras et al., 2024).**
   a. References added and discussed *(lines 360-366).*
2. **In Section 3.2 PM2.5 is evaluated relative to AirNow, a novel yet limited database. Have you considered any other dataset that may extend further back in time or over a larger geographic area in Africa?**
   a. We assessed all publicly available research-grade PM2.5 monitoring networks over Africa, but the AirNow database has the best geographic coverage we were able to find. WHO also provides surface PM2.5 data of a reliable quality (World Health Organisation, 2023), but this data is mainly for

South Africa and Senegal, and so did not cover a large enough area to be applicable to this analysis.

3. **In Section 3.3 the evaluation of seasonal dust and non-dust AOD distributions is presented in Fig. 6. The information obtained from Fig. 6 could be put on a separate table (maybe in the supplement) so the reader would know how well each CMIP6 model performed. The same could be applied to the rainfall results in Fig. 11.**
   a. These have both been added to the supplement as Table S2 and S3.
4. **In Section 3.3 non-dust AOD is evaluated altogether. Maybe an evaluation of black carbon, organic aerosols and sulphates AODs separately could shed light on the cause of poorer model performance for non-dust AOD over Africa.**
   a. This would be excellent analysis for further investigation of the roots of biases in non-dust AOD. Unfortunately the vast majority of CMIP6 models do not store BC or SO4 AOD, or bin the aerosol types separately when the model runs. Some models do store OA, but these are a small fraction of the models evaluated. We now note this in the text as part of our discussion of model structural uncertainty *(lines 695-696)*.

**Minor Comments**

**Figure 1: As AOD is unitless, the brackets "[1]" in the titles of the four colorbars can be omitted. Same goes for the colorbar titles in Figs. 4 and 5, and the Y-axis labels in Figs. 7 and 8.**

After discussion with co-authors, we have agreed to leave the labels unaltered, as the current labelling is standard for the field.

**Line 77: "Precipitation biases occur spatially" can be changed to "Precipitation biases occur both spatially".**

Changed.

**Line 83: "between the Sahara and Gulf of Guinea discussed in Cook (1999)" could be changed to "between the Sahara and the Gulf of Guinea (Cook, 1999)".**

Changed.

**Line 85: "with a focus on both regions" can be changed to "focusing on both regions".**

Changed.

**Line 99: "demonstrating" can be replaced by "presenting" or "showing".**

Changed.

**Line 101: "regional annual cycles in AOD" could be changed to "regional annual AOD cycles".**

Changed.

**Line 102: "performance of CMIP6 models over Africa".**

**Line 102: "model evaluation is performed for CMIP6 models' seasonal spatial". Please rephrase.**

Exploring the performance of CMIP6 models over Africa, the spatial distribution of rainfall by season over the whole of Africa in CMIP6 models is evaluated, as well as the representation of regional African monsoon systems.

**Table 1: It would greatly benefit this table if a column containing the measurement period for each station were added.**

Excellent suggestion, added.

**Table 1 caption: "The numbers in the final column". Also, a dot should be added at the end of this sentence.**

Corrected.

**Lines 143-144: "MODIS face, where surface brightness over deserts causes a lack of contrast between aerosol signal and the underlying surface brightness (Wagner et al., 2010), or where systematic biases are present when clouds" could be altered to "MODIS face, such as surface brightness over deserts causing a lack of contrast between aerosol signal and the underlying surface brightness (Wagner et al., 2010), or systematic biases being present when clouds".**

Changed.

**Line 148: "(non-dust AOD = total AOD − dust AOD)" could be changed to "(non-dust AOD, i.e., the total AOD minus dust AOD)".**

Changed.

**Line 175: "These datasets were concatenated".**

Changed.

**Line 176: "The list of models used is shown in Table 2, with the nominal atmosphere and ocean resolutions also shown" could be altered to "The models used in the**

study, along with their nominal atmospheric and ocean resolutions are shown in Table 2".

Changed.

**Table 2: Please check if the model reference for CAMS-CSM1-0 is correct in the following URL: http://www.climatechange.cn/EN/10.12006/j.issn.1673-1719.2019.186**

Updated.

**Figure 2 caption: The acronym NRMSE is first defined in Line 340. It should also be explained here.**

Added.

**Line 276: "analysis in shown later in Sect. 3.3" should be changed to "analysis shown in Sect. 3.3".**

Corrected.

**Line 278: Acronyms EAM and WAM are not defined until Line 490. They should be defined here instead.**

Added.

**Line 292: The word "generation" can be omitted.**

Removed.

**Figure 4 caption: "(d) intermodel standard deviation for dust AOD (shading) and wind speed".**

Changed.

**Line 381: "during JJA, with westerlies bringing in".**

Corrected.

**Lines 395-396: "over east Africa is less well represented than of that over west Africa, with the MMM failing the capture the AOD climatology" could be changed to "over east Africa is not represented as well as over west Africa, with the MMM failing to capture the AOD climatology".**

Changed.

**Line 421: "the areas with the largest intermodel spread".**

Changed.

**Line 431: "models analysed are found". For consistency with the rest of the manuscript.**

Corrected.

**Line 432: "biases in the location of rainfall, and magnitude of rainfall in regions" could be changed to "biases in the location and magnitude of rainfall over some regions".**

Changed.

**Line 434: "small areas of localised biases over". For the same reason as in Line 431.**

Changed.

**Line 444: "a ranking of the pattern correlations of each of the models evaluated for each season" could be altered to "a ranking of the pattern correlations of individual models evaluated for each season".**

Changed.

**Line 494: "individual models have pattern".**

Corrected.

**Line 519: "daily rainfall characterisation". For the same reason as in Line 431.**

Corrected.

**Lines 534-535: "from this section of the analysis".**

Changed.

**Line 541: "a much higher range" could be changed to "a wider range".**

Changed.

**Line 570: "<" should be replaced with "less than".**

Corrected.

**Line 574: "intermodel spread caused by".**

Corrected.

**Line 583: "inability to correctly model transport of aerosols from biomass burning over central Africa" can be changed to "inability to correctly simulate the transport of biomass burning aerosols over central Africa".**

Changed.

**Lines 599-602: "Therefore, these findings… region of interest.". This sentence is quite lengthy and can be confusing. I suggest rephrasing it or breaking it into two smaller ones.**

Changed to "This evaluation will inform the analysis of modelled responses in RAMIP experiments, which investigate the impacts of local and remote aerosol emission changes on Africa. The results here inform the degree of confidence that can be placed in each model's responses based on their performance for the season and region of interest."

**References RC2**

**Kalisoras, A., Georgoulias, A. K., Akritidis, D., Allen, R. J., Naik, V., Kuo, C., Szopa, S., Nabat, P., Olivié, D., van Noije, T., Le Sager, P., Neubauer, D., Oshima, N., Mulcahy, J., Horowitz, L. W., and Zanis, P.: Decomposing the effective radiative forcing of anthropogenic aerosols based on CMIP6 Earth system models, Atmospheric Chemistry and Physics, 24, 7837–7872, https://doi.org/10.5194/acp-24-7837-2024, 2024.**

**Thornhill, G. D., Collins, W. J., Kramer, R. J., Olivié, D., Skeie, R. B., O'Connor, F. M., Abraham, N. L., Checa-Garcia, R., Bauer, S. E., Deushi, M., Emmons, L. K., Forster, P. M., Horowitz, L. W., Johnson, B., Keeble, J., Lamarque, J.-F., Michou, M., Mills, M. J., Mulcahy, J. P., Myhre, G., Nabat, P., Naik, V., Oshima, N., Schulz, M., Smith, C. J., Takemura, T., Tilmes, S., Wu, T., Zeng, G., and Zhang, J.: Effective radiative forcing from emissions of reactive gases and aerosols – a multi-model comparison, Atmos. Chem. Phys., 21, 853–874, https://doi.org/10.5194/acp-21-853-2021, 2021.**

**Zelinka, M. D., Smith, C. J., Qin, Y., and Taylor, K. E.: Comparison of methods to estimate aerosol effective radiative forcings in climate models, Atmospheric Chemistry and Physics, 23, 8879–8898, https://doi.org/10.5194/acp-23-8879-2023, 2023.**

Other notes:
- A bug for plotting the standard deviation panel in Figures 4, 5, and 10 was found, and has been corrected. This did not affect the text associated, and has not affected the conclusions of the paper.
- The EC-Earth3 model family was missing from our list of models - this has been corrected.

- The AirNow dataset has been terminated since the original submission of this paper. This is remarked on in the conclusion *(line 642).*

References AC

Guo, L., L. J. Wilcox, M. Bollasina, S. T. Turnock, M. T. Lund, and L. Zhang, 2021: Competing effects of aerosol reductions and circulation changes for future improvements in Beijing haze. Atmos. Chem. Phys., 21, 15299–15308, https://doi.org/10.5194/acp-21-15299-2021.

Shindell, D. T., A. Voulgarakis, G. Faluvegi, and G. Milly, 2012: Precipitation response to regional radiative forcing. Atmos. Chem. Phys., 12, 6969–6982, https://doi.org/10.5194/acp-12-6969-2012.
Shonk, J. K. P., A. G. Turner, A. Chevuturi, L. J. Wilcox, A. J. Dittus, and E. Hawkins, 2020: Uncertainty in aerosol radiative forcing impacts the simulated global monsoon in the 20th century. Atmos. Chem. Phys., 20, 14903–14915, https://doi.org/10.5194/acp-20-14903-2020.

Turnock, S. T., and Coauthors, 2020: Historical and future changes in air pollutants from CMIP6 models. Atmos. Chem. Phys., 20, 14547–14579, https://doi.org/10.5194/acp-20-14547-2020.

World Health Organization. (2023). WHO ambient air quality database, 2022 update: status report. Genève, Switzerland: World Health Organization.

Williams, A. I. L., P. Stier, G. Dagan, and D. Watson-Parris, 2022: Strong control of effective radiative forcing by the spatial pattern of absorbing aerosol. Nat. Clim. Chang., 12, 735–742, https://doi.org/10.1038/s41558-022-01415-4.

Zhao, A., C. L. Ryder, and L. J. Wilcox, 2022: How well do the CMIP6 models simulate dust aerosols? Atmos. Chem. Phys., 22, 2095–2119, https://doi.org/10.5194/acp-22-2095-2022.

---

## Author Response (AR2)

We thank the editor for their helpful comments for improving our manuscript. The editor's comments are shown in **bold**, with responses below in standard font, and lines noted in the latexdiff manuscript where relevant changes have been made in *italics.*

**1) The calculation of PM2.5 from models with vastly different size distributions and representations of super coarse dust is quite uncertain. Please caveat your description of this calculation "In cases where the CMIP6 models do not provide PM2.5 directly, we calculate it from speciated mass-mixing ratios at the surface, following Turnock et al. (2020): PM2.5 = BC + SO4 + OA + (0.25 × SS) + (0.1 × DU)" accordingly and discuss potential errors.**
Discussion added *(lines 247- 264)*, reflecting:
- Inconsistency between models, i.e. having to calculate PM2.5 for some, some having PM2.5 calculated from size distributions, and others performing postprocessing to calculate PM2.5 using methods similar to above to output PM2.5, makes comparison harder.
    - Models for which PM2.5 could be calculated using equation above were included to improve range of model behaviour discussed.
- Calculation for PM2.5 depends on assumptions about size distribution of individual aerosol species.
- Biases in concentrations for MMRs cause biases in PM2.5.
- Some of Turnock et al (2020)'s evaluation:
    - Areas of strong dust emissions are associated with larger diversity across CMIP6 models, noting the Saharan region as an area of high uncertainty.
    - Outside of these dusty regions CMIP6 models are relatively similar in their simulation of PM2.5 concentrations.
    - Small model bias in PM2.5 concentrations across most regions, excluding oceans where there is higher model diversity,
        - Africa not included for that part of analysis.
    - Generally associated with underestimation of PM2.5 in most regions, likely because the PM2.5 calculation excludes nitrate aerosols.
- Largest source of model diversity for the majority of stations over Africa is differences in dust.
- Another important contributor to model diversity in calculated PM2.5 at coastal locations will be SS.
- No consistent negative or positive bias for models with calculated PM2.5 when compared to observations or models for PM2.5 output available.
    - But outputs from some models for PM2.5 may use similar postprocessing methods to produce PM2.5, so there may not be a clear distinction between models with and without PM2.5 outputs available.

**2) Likewise, dust AOD is not directly observationally constrained in CAMS, so it would be good to briefly discuss related uncertainties.**

Discussion added *(lines 167-180)*, noting that:
- Necessary to understand that speciated AOD evaluation is done in the context of substantial reanalysis uncertainty and disagreement.
  - Vogel et al (2022) has conducted a review comparing uncertainty in AOD across reanalysis, satellite retrieval, and CMIP model dataset, finding regional AOD from MMMs often fall outside of the range from satellite products, e.g., over North Africa.
  - However North Africa can be seen to be one of the regions of highest uncertainty in Vogel et al (2022).
  - CAMS performs well over Africa in this study, with weaker performance found over east Asia (and localized positive biases associated with volcanic activity Saturno et al (2018)).
- Accurate representation of dust AOD in reanalysis relies on simulation of correct concentration of dust relative to other aerosol species. Aerosol speciation is better represented in locations dominated by dust (e.g. the Sahara), likely to be less well represented in regions where different aerosol species coexist (e.g. northern India, with mixed dust, smoke, and anthropogenic aerosol).
- Dust processes in CAMS and MERRA2 are model-dependent and have associated uncertainties (Xian et al. (2020), Zhao et al. (2022)). Comparisons between models and the reanalyses presented should be interpreted with some caution.
- Dust is distinguished from non-dust aerosols based on fine-mode fraction, Ångström exponent, and single-scattering albedo (Song et al (2021), Inness et al (2019))

References:
Vogel, A., G. Alessa, R. Scheele, L. Weber, O. Dubovik, P. North, and S. Fiedler, 2022: Uncertainty in Aerosol Optical Depth From Modern Aerosol‐Climate Models, Reanalyses, and Satellite Products. JGR Atmospheres, 127, https://doi.org/10.1029/2021jd035483.

Saturno, J., and Coauthors, 2018: African volcanic emissions influencing atmospheric aerosols over  the Amazon rain forest. Atmos. Chem. Phys., 18, 10391–10405, https://doi.org/10.5194/acp-18-10391-2018.

Garrigues, S., and Coauthors, 2022: Monitoring multiple satellite aerosol optical depth (AOD) products within the Copernicus Atmosphere Monitoring Service (CAMS) data assimilation system. Atmos. Chem. Phys., 22, 14657–14692, https://doi.org/10.5194/acp-22-14657-2022.

Song, Q., Z. Zhang, H. Yu, P. Ginoux, and J. Shen, 2021: Global dust optical depth climatology derived from CALIOP and MODIS aerosol retrievals on decadal timescales: regional and interannual variability. Atmos. Chem. Phys., 21, 13369–13395, https://doi.org/10.5194/acp-21-13369-2021.

Inness, A., and Coauthors, 2019: The CAMS reanalysis of atmospheric composition. Atmos. Chem. Phys., 19, 3515–3556, https://doi.org/10.5194/acp-19-3515-2019.

Xian, P., P. J. Klotzbach, J. P. Dunion, M. A. Janiga, J. S. Reid, P. R. Colarco, and Z. Kipling, 2020: Revisiting the relationship between Atlantic dust and tropical cyclone activity using aerosol optical depth reanalyses: 2003–2018. Atmos. Chem. Phys., 20, 15357–15378, https://doi.org/10.5194/acp-20-15357-2020.

Zhao, A., C. L. Ryder, and L. J. Wilcox, 2022: How well do the CMIP6 models simulate dust aerosols? Atmos. Chem. Phys., 22, 2095–2119, https://doi.org/10.5194/acp-22-2095-2022.